# Strategic deployment of solar photovoltaics for achieving self-sufficiency in Europe throughout the energy transition

Parisa Rahdan [1,2] ✉, Elisabeth Zeyen [3] & Marta Victoria [1,2,4]

Transition pathways for Europe to achieve carbon neutrality emphasize the need for a massive deployment of solar and wind energy. Global cost optimization would lead to installing most of the renewable capacity in a few resource-rich countries, but policy decisions could prioritize other factors. We investigate the effect of energy independence on Europe's energy system design. We show that self-sufficiency constraints lead to a more equitable distribution of costs and installed capacities across Europe. However, countries that typically depend on energy imports face cost increases of up to 150% to ensure that they cover their demand on an annual basis. Self-sufficiency particularly favors solar photovoltaic energy, and with declining PV module prices, alternative configurations like inverter dimensioning and horizontal tracking are beneficial enough to be part of the optimal solution for many countries. Moreover, we find that very high solar and wind annual installation rates are required, but they seem feasible considering recent historical trends.

As Europe advances in its green transition, it is crucial to harmonize national policies to achieve a net-zero emissions system by 2050. This study addresses three major research gaps in macro-energy system planning: the impact of countries' energy self-sufficiency on transition pathways, the role of new solar photovoltaic (PV) configurations, and the feasibility of the required high growth rates for wind and solar PV based on historical trends.

Energy self-sufficiency is the capability to satisfy energy needs without depending on others. Although collaboration will be necessary to achieve a renewable European energy system with minimal costs[1], most European countries would like to attain a certain degree of self-sufficiency for energy security. This inclination has been intensified by the gas crisis triggered by Russia's invasion of Ukraine, re-opening the discussion in several countries on the convenience of increasing the use of nuclear or even fossil fuels[2–7] to lessen their reliance on energy imports[8]. Previous works[9–13] have investigated the role of self-sufficient generation for different European countries, but these analyses were either limited to the power sector, one region, or just one specific point in time, missing the potential path-dependency observed when

modeling transition paths. To address these gaps, in this work, we examine the effect of self-sufficiency on the transition to a climate-neutral sector-coupled European energy system.

Solar PV electricity is highlighted as the most cost-effective mitigation investment globally[14], and its deployment increases when pursuing regional equity or reducing gas imports for Europe[9,15–18]. Distributed PV, installed on rooftops or parking lots, can also increase self-sufficiency in highly populated regions[19]. Considering the importance of solar PV both for the transition and in order to achieve self-sufficiency, we aim to improve the representation of solar PV in macro-energy systems models. To do this, we model several emerging PV configurations that are often overlooked in studies but could be cost-efficient due to the dramatic cost reduction experienced by PV modules, driven by rapid learning curves[20]. The first one is inverter dimensioning, meaning the inverter converting PV power from direct current (DC) to alternating current (AC) is undersized on the AC side since PV modules rarely generate power at full capacity (with standard high solar irradiation)[21,22]. This practice lowers costs, even if it leads to some power curtailment during very sunny hours, and is becoming

[1]Department of Mechanical and Production Engineering and iCLIMATE Interdisciplinary Centre for Climate Change, Aarhus University, Aarhus, Denmark. [2]Department of Wind and Energy Systems, Technical University of Denmark, Lyngby, Denmark. [3]Department of Digital Transformation in Energy Systems, Technische Universität Berlin, Berlin, Germany. [4]Novo Nordisk Foundation CO2 Research Center, Aarhus, Denmark. ✉e-mail: parra@dtu.dk

more advantageous as PV module costs decrease faster than AC components like inverter and grid connection. Second, horizontal single-axis tracking (HSAT), where PV modules rotate from east in the morning to west in the evening, extends solar generation hours and is already cost-efficient, holding a 60% market share in new utility PV installations in 2023–2024[23]. With the exception of studies by research groups of Breyer[24,25] and Blakers[26], HSAT is often excluded from macro-energy system models. Third, inexpensive PV modules enable alternative system designs like the delta configuration, which comprises triangular rows of modules facing east and west. This non-optimal orientation results in lower annual electricity generation per DC capacity, but matches the daily profile of HSAT without moving parts, increasing self-consumption and attaining higher energy yield per area[27–30]. Other advantages of delta configuration are reduced wind loads and structural weight for rooftop systems[31], easy installation for the many homes with east or west-facing rooftops[32], and possible integration with agriculture as vertical east-west bifacial modules[33,34]. In addition to the question of cost-effectiveness, we explore whether these configurations support self-sufficiency for a carbon-neutral Europe.

Installing large-scale capacities of wind and solar PV has been shown to be a cost-effective strategy to achieve a carbon-neutral Europe[35–39] and increase energy security[17]. However, the large required installation rates for wind and solar have been questioned to be feasible[40,41] or possible by social acceptance issues[42–44]. We address here the question of whether ensuring a certain degree of self-sufficiency while decarbonizing the different European countries modifies the wind and solar installation rates, and whether these rates can be considered achievable when looking at historical rates.

This work introduces three main contributions: (1) modeling a Paris Agreement-compatible transition for self-sufficient interconnected European countries, (2) examining emerging PV configurations, and (3) assessing required installation rates for wind and solar, contextualized with historical data. We model the European sector-coupled networked energy system from 2025 to 2050 with 5-year steps, under a carbon budget corresponding to 1.7 °C temperature increase and imposing carbon neutrality by 2050, with a network comprising 37 nodes, using 370 regions to represent wind and solar resources, and a time-resolution of 2-h for a full year. The results indicate that self-sufficiency minimally impacts total costs but promotes fairer capacity distribution. However, costs increase up to 150% for net-importer countries by 2050. High-value synthetic fuels are produced in renewable resource-rich countries, and hydrogen is traded extensively. Alternative PV configurations also aid self-sufficiency by reducing costs and extending generation. Ultimately, growth rates for both wind and solar require higher ambitions in many European countries, but the trends of recent years show that they are achievable. The upcoming section delves into the main results, beginning by discussing a baseline transition-to-net-zero scenario, followed by describing the impacts of introducing a self-sufficiency constraint and alternative PV configurations. Based on the results, we make recommendations to energy modelers and policymakers. Lastly, the "Methods" section outlines the mathematical formulation of the self-sufficiency constraint and the key modeling assumptions, in particular regarding solar PV.

## Results and discussion

### A climate-neutral Europe with energy-independent countries

Our analysis proceeds as follows: First, we examine the impact of implementing self-sufficiency on the energy system at European and country-specific levels. After establishing the critical role of solar PV in achieving self-sufficiency, we proceed to the second part, where we use an overnight scenario to evaluate various new solar configurations. This analysis identifies the cost-efficient configurations that should be incorporated into the model. Third, we revisit the self-sufficiency

experiment using these newly selected configurations to assess their impact on the system, both with and without the self-sufficiency constraint. Finally, we conclude by analyzing the transition pathways for wind and solar energy across different countries under various scenarios, evaluating whether the required ambitions are achievable based on historical growth rates.

We use an open-source model to optimize the capacity and dispatch of all system elements, aiming to minimize total system costs while adhering to defined constraints (refer to "Methods" for a more detailed description). Covering the entire ENTSO-E[45] area, we use a spatial resolution of 37 nodes with 370 regions for renewable potential estimation and capture a one-year period with 2-h time steps. The model integrates electricity generation, including solar, onshore and offshore wind, hydropower, nuclear, methane gas, and coal, storage technologies such as batteries, and transmission grid, as well as a simplified distribution grid. Additionally, it incorporates the heating, land transport, aviation, shipping, industry (including industrial feedstock), and agriculture sectors, considering their specific demands and incorporating relevant technologies like heat pumps, electric vehicles, and industrial processes (see Supplementary Note 1 for more details).

### Assessing system-wide impacts of self-sufficiency

We model the energy transition in Europe from 2025 to 2050 in 5 years time steps, using a myopic approach and a carbon budget corresponding to 1.7 °C temperature increase (see Supplementary Fig. 3). First, we explore the transition path without any self-sufficiency requirements and compare it with the results when adding a self-sufficiency target, which requires that the self-sufficiency coefficient for every country (see Equation (2) in "Methods") reaches 60%, 80%, and 100% in 2030, 2040, and 2050, respectively. Total system cost composition for both transition paths is shown in Fig. 1a. Implementing the self-sufficiency constraint increases total system cost by 5.1% for the last investment period, and shows an average increase of 2.1% during the whole transition. The small total system cost increase in the case of self-sufficiency indicates that the cost-optimal solution space is very flat (as explored in several studies[16,18,37,46]), and self-sufficiency could be achieved with small additional expenses.

However, on a national level, the system costs can increase strongly, with some countries experiencing up to a 150% rise (Fig. 1b). Under the self-sufficiency constraint, renewable generation and electrolysis capacities are moved from countries that previously were net exporters, such as Spain or Denmark, to previous net-importing countries, such as Belgium, the Netherlands, or Germany. These countries tend to be net importers due to their high industry demand and population density. For Belgium, which experiences the highest cost increase, the surge is largely driven by the need for additional nuclear capacity when wind and solar reach their assumed potential, which, despite being relatively small, incurs substantial costs due to the high expense of the technology (see Supplementary Fig. 22).

Implementing self-sufficiency results in a more balanced exchange of energy carriers between countries (Fig. 1c). In the model, energy can be exchanged among countries in the form of electricity by AC and DC transmission lines, biomass by trucks, hydrogen and gas by pipelines, and oil and methanol assuming negligible transport costs and no transmission bottlenecks. Electricity network expansion is limited to 10% of today's capacity, and only 13% of the total energy transported between countries is through AC and DC lines. A sensitivity analysis with higher transmission expansion is discussed in Supplementary Note 5. Gas exchange between countries reduces after 2025 (see Supplementary Figs. 19–22), since gas demand is reduced by electrified heating due to the carbon budget constraint. Consequently, hydrogen, synthetic oil, and synthetic methanol are the most relevant energy imports and exports for the 2050 net-zero emissions system, as they are used in the transport and industry sectors. Hydrogen is mostly produced from electrolysis, and synthetic fuels are produced by

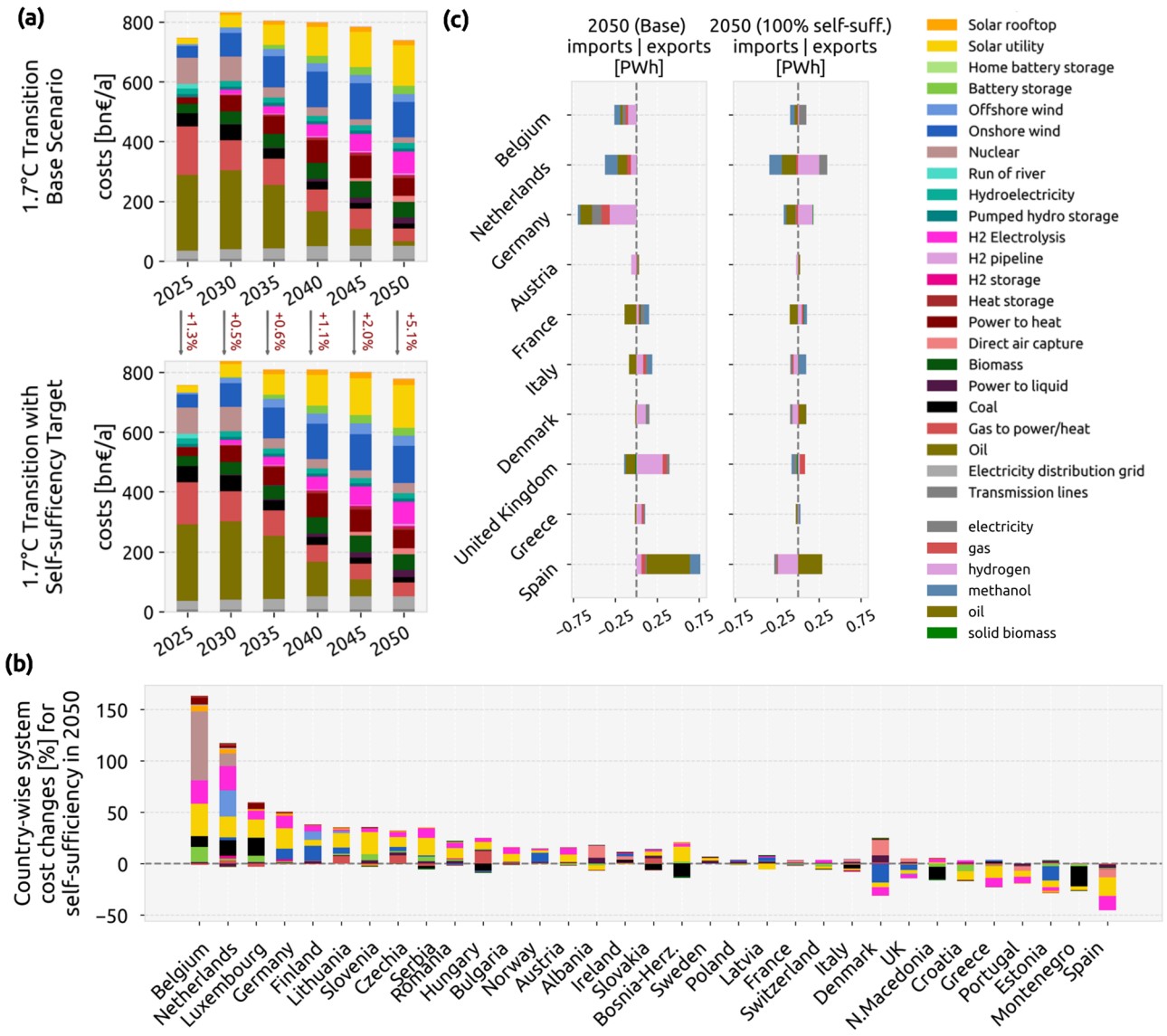

**Fig. 1 | Effect of the self-sufficiency constraint on the Energy system.** Changes in **a** total system costs during the transition under a 1.7 °C temperature increase target with and without self-sufficiency, **b** country-specific system costs (%) for year 2050 for the 100% self-sufficient scenario relative to the base scenario, showing what components are responsible for the cost increase (%) in each country, and **c** energy imports (negative) and energy exports (positive) of each country for year 2050 for the 100% self-sufficient scenario and base scenario.

combining electrolytic hydrogen and captured $CO_2$ using Fischer-Tropsch reaction and methanolisation units.

In the base transition scenario, oil and methanol are produced either in countries with abundant renewable resources, like Spain, or in high-demand countries, like Germany, which import the necessary hydrogen and electricity from nearby countries with strong renewable potential, such as Denmark and the UK. However, the self-sufficiency constraint results in spreading the production of synthetic oil, and methanol to a lesser degree, to all countries (Fig. 1c and Supplementary Fig. 22). The share of oil and methanol from total exports is still high in countries such as Spain, the UK, and France. Germany and several other countries import their entire methanol demand, and the Netherlands and Portugal rely on imports for over 90% of their oil demand (see Supplementary Figs. 19–22). This is because the self-sufficiency constraint is imposed on the sum of all energy carriers (Equation (3)), which makes the system benefit from producing high-value energy carriers in countries with good renewable resources. Therefore, European cooperation is still necessary for achieving self-sufficiency with minimal costs.

Solar PV and wind energy become the cornerstone of the transformed energy system, with solar PV being crucial for achieving self-sufficiency. By 2050, 5.1 TW of solar and 1.3 TW of onshore and off-shore wind capacity will be installed across Europe (see Supplementary Fig. 13), taking up 57% and 36% of the electricity generation, respectively. This requires approximately 2.3% of available land for solar PV and 3.7% for wind, equivalent to about 4.5% of Europe's total land area. This land use aligns with very conservative estimates of suitable land for renewable energy installations that account for environmental, agricultural, biodiversity, and social constraints[47]. However, achieving these targets may be more challenging for certain countries.

Solar PV plays a more prominent role in our scenarios than previous similar studies[35,48], because we have included inverter dimensioning in the model. Including inverter dimensioning effectively reduces the investment cost of utility solar PV in our model by 15.8% relative to a PV system with the same DC and AC capacity. This decrease in costs causes the system to have a 1.7 ratio for solar generation to wind generation, which is more than twice what was observed in previous similar studies[48]. While this ratio seems high,

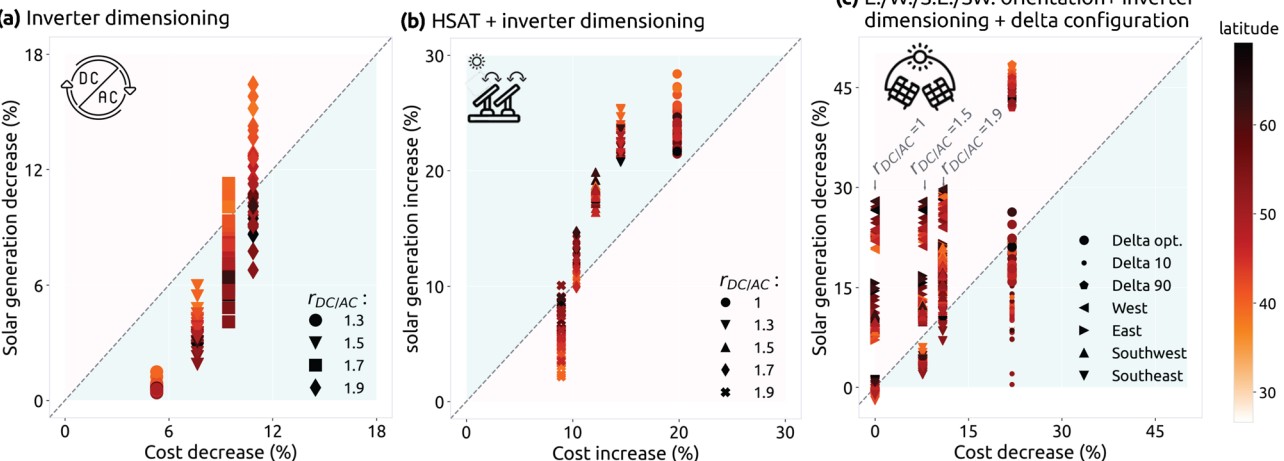

**Fig. 2 | Cost-efficiency of alternative solar configurations.** Changes in annual electricity generation vs. cost for the different PV configurations compared to a South-facing module with inverter ratio of 1 for: **a** higher inverter ratios (ratio of installed solar DC capacity to AC power rating of the inverter shown as $r_{DC/AC}$), **b** horizontal single-axis tracking (HSAT) with inverter dimensioning, and **c** delta configuration, with 10°/optimal/90° inclination, plus east/west/southeast/southwest-facing modules with optimal inclination and inverter dimensioning (The optimal inclination is the best tilt angle for south-facing modules in each country). The blue areas show configurations that are considered cost-efficient, and the further a point is from the diagonal line, the higher its cost benefit or detriment.

studies incorporating recent drops in solar PV costs have shown even higher ratios, up to 1.9[49]. However, as this ratio changes with different cost assumptions, we will focus on robust results in this section, such as the role of methanol and hydrogen. Refer to Supplementary Notes 3 and 4 for more details.

Notably, solar PV is the only technology whose capacity consistently increases across all countries that need to boost local generation to achieve self-sufficiency (Fig. 1b). Given the significant deployment of both utility-scale and distributed solar capacities, we will explore the potential of alternative solar technologies to improve energy independence in the subsequent discussion.

### Solar PV tracking and inverter dimensioning are economic

As our model uses 370 regions to assess solar resources, adding each new solar configuration increases computational complexity. Therefore, we perform two simplified, highly renewable overnight scenarios to determine which configurations merit further investigation under the transition scenario. In both scenarios, 19 alternative solar PV configurations are evaluated, which were chosen based on an initial cost-benefit analysis (Fig. 2).

For utility-scale PV generation connected to the high-voltage grid, we evaluate three configurations: (1) south-facing and (2) horizontal single-axis tracking (HSAT), both with different inverter ratios, and (3) delta configuration with 10° inclination and 1.5 inverter ratio. For rooftop PV systems connected to the low-voltage grid, we consider the same configurations as utility-scale except HSAT, and instead add southeast and southwest-facing modules with an inverter ratio of 1.9. Additional details regarding the costs, land use, and generation profiles of these new solar configurations are provided in the "Methods" and Supplementary Notes 3 and 4.

Different inverter ratios and HSAT are selected in both overnight scenarios (see Supplementary Fig. 14). Delta configuration is only selected for the scenario with a 100% self-sufficiency target. Southwest and southeast are not selected in any case. The reason for a configuration to be selected is mainly based on the solar generation gains vs. the increase in costs (or vice-versa) in every country (Fig. 2). First, for all countries in Europe, the cost decrease of selecting a DC/AC ratio of 1.3 or 1.5 is higher than the reduction in annual electricity generation (blue area of the plot in Fig. 2a). This confirms that solar generation at nominal DC capacity is infrequent enough to justify oversizing the PV modules or DC capacity of the plant. For

example, a DC/AC ratio of 1.3 results in annual energy loss below 3% for every country, but higher ratios like 1.7 and 1.9 are not beneficial for Southern Europe due to higher losses. However, cost vs. energy is not the single deciding factor for selection of a configuration, as demand at peak solar production hours, generation from other renewable sources, and the available transmission or storage capacity will also impact the selection.

Second, Fig. 2b shows all DC/AC ratios except 1.9 are cost-efficient to use with HSAT for all countries in Europe. Interestingly, Southern countries gain the highest increase in energy production for a low DC/AC ratio, but ratios higher than 1.5 are more appropriate for Northern European countries since the number of hours at high DC generation with HSAT is still low (see Supplementary Fig. 4).

Third, east, west, southeast, and southwest-facing modules are generally not cost-efficient or much less cost-efficient than configurations like HSAT. Delta configuration reduces the annual generation (relative to the installed DC capacity, as discussed in Supplementary Note 3) but also the cost. Delta configuration with 10° inclination angle can be beneficial for any European country (Fig. 2c), but optimal tilt (same as south-facing) is only attractive for higher latitudes where the Sun is low during most of the year.

Despite the results from the simplified evaluation in Fig. 2c, when alternative orientation angles or delta configurations are evaluated in the overnight cost optimization, they are not selected (see Supplementary Fig. 14). This is because HSAT is more cost-efficient than these configurations for increasing PV generation in the early morning or late afternoon. These configurations can, however, still be useful for single consumers who would benefit from them depending on their demand and the electricity prices in the region.

Overall, the results here indicate that consideration of inverter dimensioning when modeling solar PV is essential, as for both south-facing fixed modules and HSAT, almost the entire capacity is paired with an inverter ratio higher than 1.

### Alternative solar PV configurations support self-sufficiency

We model again the transition paths including the alternative solar configurations identified as cost-effective in the previous overnight optimization exercise. These include south-oriented configuration with inverter ratios of 1.5 and 1.7 (both for utility-scale and distributed systems), HSAT with inverter ratios of 1.3 and 1.5, and delta configuration with 10° inclination and inverter ratio of 1.5. For a detailed

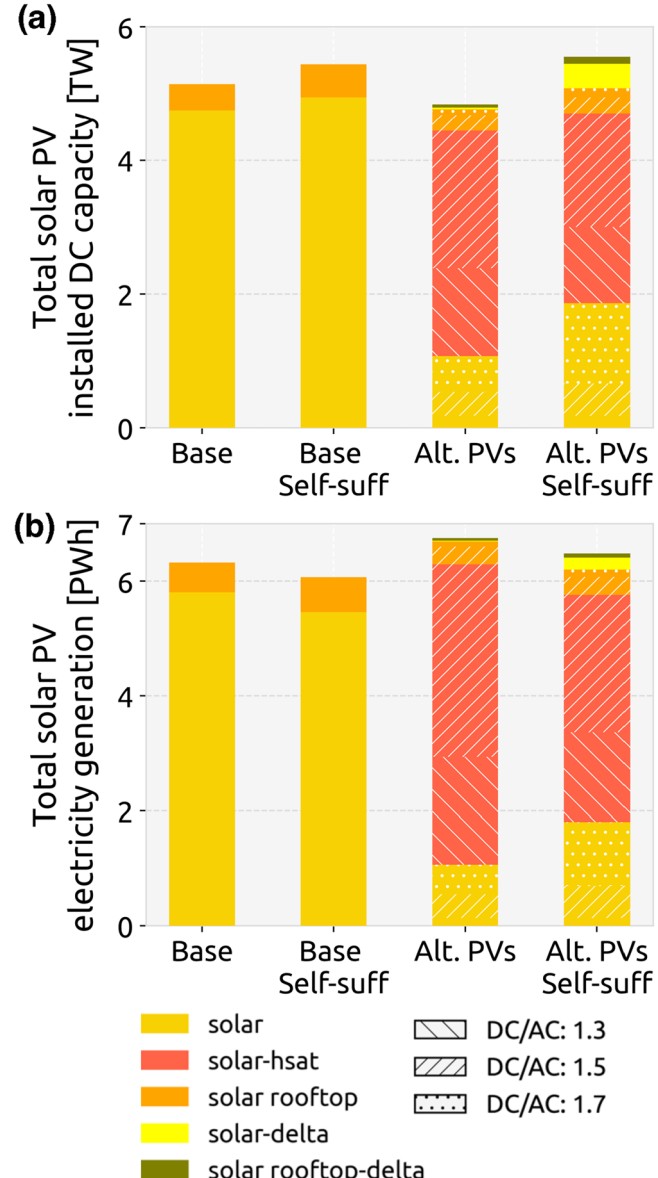

**Fig. 3 | Capacity and generation mix for solar PV. a** Cumulative installed DC capacity and **b** annual electricity generation from different solar configurations for all transition scenarios in 2050.

explanation of the cost assumptions related to inverter dimensioning and delta configurations, see Supplementary Note 3.

Including the self-sufficiency target leads to a lower Europe-average capacity factor of solar PV for both the base scenario and the Alternative PV scenario, since solar PV is installed in less optimal locations (Figs. 3, 4). Therefore, despite higher installed capacity, total solar generation under the self-sufficiency target decreases (Fig. 3) and is compensated by higher wind generation in the system (see Supplementary Fig. 13). Addition of alternative PV configuration results in the installation of large shares of HSAT in many countries. For the Alternative PV scenario, the higher capacity factors attained by HSAT help achieve higher electricity generation from solar PV compared to the base scenario, despite a lower DC capacity being installed in the system. Under the self-sufficiency constraint, a higher capacity of solar PV is installed in the system when alternative configurations are available. Despite this higher capacity, the total system cost for the transition with self-sufficiency target is lowered by an average of 1.4%

when alternative solar configurations are added (see Supplementary Fig. 12).

HSAT is notably installed in the most southern and most northern countries, with average higher DC/AC ratios in southern European countries (Fig. 4). These countries benefit from the longer solar production hours that HSAT provides, but do not need the extra generation during peak generation hours around noon (see Supplementary Fig. 15), hence the prevalence of 1.5 DC/AC ratio even though 1.3 is more cost-efficient for most countries (Fig. 2). Adding a self-sufficiency target to the transition has two noticeable impacts when it comes to solar PV installations. First, the delta configuration represents a large share of solar generation in Belgium and the Netherlands, and a small share in Poland and Germany. This is mostly driven by the land availability constraints in those regions since the delta configuration enables reduced land use for the same capacity. Second, there is a shift from HSAT with a DC/AC ratio of 1.5 to 1.3 for countries like Spain, France, and Switzerland, while Germany and Belgium experience this shift in reverse, installing more HSAT with a DC/AC ratio of 1.5. This means that the system still does not require extra generation at noon, and is shifting the capacities previously installed in Spain to other countries. The capacity shifts that happen under self-sufficiency are quite significant in some countries. We conclude the results in the next section by discussing whether these targets are realistically attainable based on past performance and deployment trends.

**Large installation rates are within reach**
Now we turn our attention to how the transition scenarios play out for each country. We compare the historical capacity deployments of solar PV and wind (both onshore and offshore) using data from IRENA[50], with what is needed from now till 2050. The transition paths of most countries are similar for all scenarios, but the self-sufficiency requirement has a noticeable impact on certain countries (Fig. 5 and Supplementary Fig. 27). Countries that were previously net importers, such as Germany and the Netherlands, increase their cumulative solar capacity, and the opposite happens in previous net exporters. Including alternative solar configurations could result in both an increase or a decrease in PV capacity, but the change is usually much smaller compared to the impact of the self-sufficiency requirement. One group of countries, such as Italy, Portugal, Sweden, and Austria, replaces their static PV capacity with HSAT to increase solar generation. The other group, including Spain, Denmark, Belgium, and the Netherlands, takes advantage of the more cost-efficient configurations and installs more solar PV to reach self-sufficiency with lower costs.

In light of historical data, the required cumulative PV capacity and annual installation rates for solar PV and wind technologies are very challenging for most countries. Nevertheless, the acceleration of installation rates in recent years is closing the gap between what would be needed and what has been shown to be possible. In previous work, Cherp and co-authors[40] raised concerns regarding the feasibility of the required installation rates based on historical data. For solar PV, they found a maximum annual rate of 0.6% (interquartile range of 0.4–0.9%) of the total electricity supply. However, their historical analysis is based on fitting S-curves to data up to 2019, a period in which wind and solar installations were mostly driven by policy-support measurements such as feed-in tariffs, especially until 2015. Figure 5 (inset figures) indicates that, for several European countries, after the end-of-policy-support stagnation, the historical data shows a second period in which capacity ramps up quickly pushed by technology competitiveness, reaching a maximum annual rate up to 3% of annual electricity generation (see Supplementary Fig. 31). Still, under the self-sufficiency constraint many countries must reach installation rates up to five times higher than the maximum they have achieved so far to be able to meet their targets (see Supplementary Fig. 28).

The same comparison of historical and needed capacities for onshore and offshore wind (Fig. 6 and Supplementary Fig. 29)

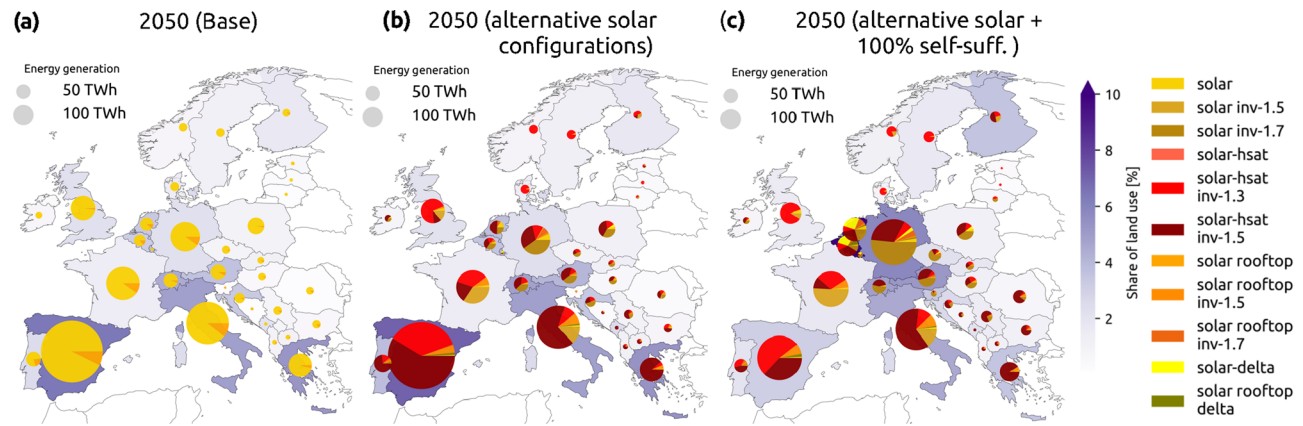

**Fig. 4 | Spatial distribution of Solar PV.** Solar generation map by configuration and regional land use (% of available land) of solar PV technologies for the year 2050 for **a** base transition, **b** transition with selected alternative solar configurations, and **c** transition with selected alternative solar configurations under a 100% self-sufficiency target.

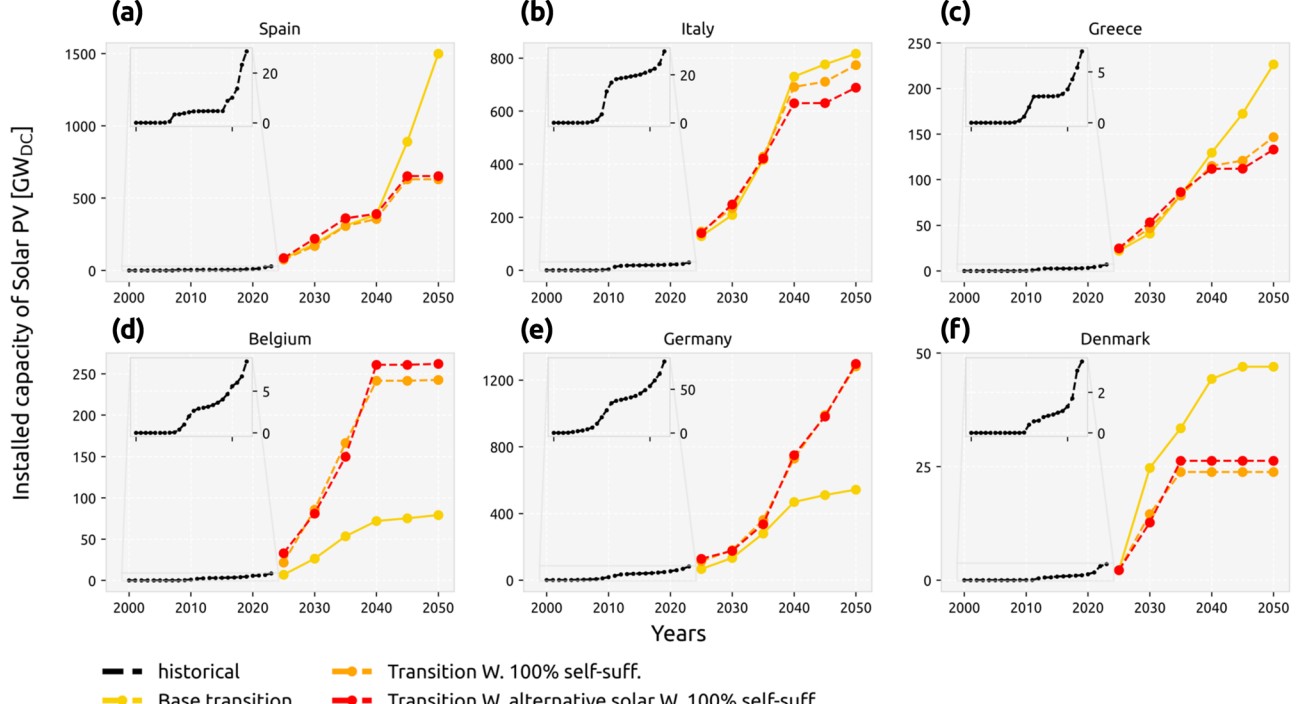

**Fig. 5 | Solar PV historical and future installation rates.** Historical (data from IRENA[50]) and modeled cumulative installed capacity of solar PV for **a** Spain, **b** Italy, **c** Greece, **d** Belgium, **e** Germany, and **f** Denmark under base transition, transition with a 100% self-sufficiency target, and transition with selected alternative solar configurations under a 100% self-sufficiency target. The insert in each figure shows a zoom on the historical years. Refer to Supplementary Figs. 27 and 28 for other countries and the required installation rates.

indicates that many European countries are on a reasonable path for meeting their targets by 2050, but still higher ambitions are necessary for countries such as Germany and Belgium. There is no dominant pattern regarding self-sufficiency targets and installation rates for wind, which can be attributed to wind resource and wind generation being non-uniformly distributed spatially compared to solar. However, many countries such as Spain, Germany, and Greece show a reduction in wind capacity when alternative solar configurations are available (Fig. 6). Overall, most countries are in a better position to meet the required wind capacities than solar PV, although countries such as the UK and Denmark still need to significantly ramp up installations in the near future.

For both wind and solar PV, growth rates might seem high compared to early historical rates, but costs are expected to continue decreasing in the future, which will enable fast growth[51–53]. Other factors that could accelerate growth are the rising social cost of carbon, which could in turn drive policies that increase carbon price in EU-ETS market[54–56], and re-emergence of local cooperatives to gain public acceptance[57,58]. However, significant barriers remain, including grid congestion, slow permitting processes, and low social acceptance[59]. Overcoming these obstacles through infrastructure upgrades, streamlined regulations, and community engagement will be essential for scaling up wind and solar PV installations.

**Limitations and further work**

Some limitations of this study, such as limiting transmission expansion to 10% of current volumes or representing wind generation using only one onshore and one offshore turbine type, are discussed in the

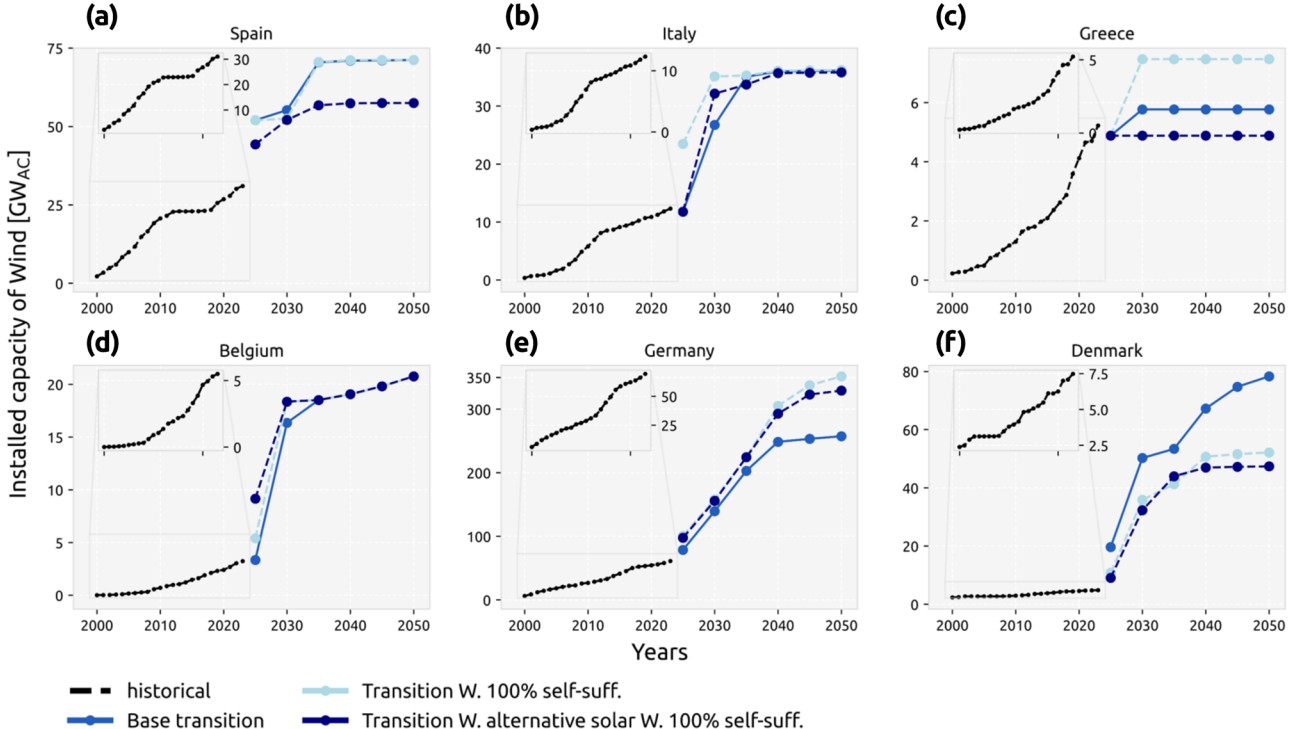

**Fig. 6 | Wind historical and future installation rates.** Historical (data from IRENA[50]) and future cumulative installed capacity of onshore and offshore wind for **a** Spain, **b** Italy, **c** Greece, **d** Belgium, **e** Germany, and **f** Denmark modeled under base transition, transition with a 100% self-sufficiency target, and transition with selected alternative solar configurations under a 100% self-sufficiency target. Refer to Supplementary Figs. 29 and 30 for other countries and the required installation rates.

Supplementary Note 5. Overall, while varying assumptions may alter the system's structure in terms of installed capacity and generation, the core findings regarding the cost-efficiency of new solar configurations and the changes needed to achieve self-sufficiency remain robust. Future research could explore the impact of extreme weather events on self-sufficiency requirements for individual countries or assess the feasibility of imposing an annual self-sufficiency constraint for a shorter time period (e.g., weekly or daily).

### Final remarks and policy recommendations

We investigate transition paths for European countries to achieve carbon neutrality by mid-century while simultaneously becoming self-sufficient. The findings show that overall system costs increase by just 2.1% under a self-sufficiency constraint, but costs can rise up to 150% for countries that were net-importers in an unconstrained scenario. Self-sufficiency also promotes a more uniform land use distribution among countries, reducing potential social acceptance issues. Therefore, pursuing energy security alongside carbon neutrality in Europe could be achieved with limited total cost increases but significant national disparities.

Solar PV is projected to contribute the largest share of electricity generation (57%), with further capacity expansions necessary under self-sufficiency due to its low cost and widespread resource availability. From a system perspective, horizontal single-axis tracking (HSAT) is favored as it extends solar generation hours, making it suitable for countries aiming for self-sufficiency. Additionally, using lower inverter capacities than the DC capacity of solar panels is cost-effective, with DC/AC ratios of 1.7 and 1.5 commonly chosen for Central and Southern Europe. The findings suggest revisions in macro-energy models, for which we recommend: (1) the inclusion of HSAT, (2) proper representation of inverter sizing, and (3) separate modeling for distributed PV systems versus utility-scale

plants[19]. In contrast, non-optimal solar orientations or delta configuration can be excluded since their value regarding displacing electricity generation in time diminishes when the system includes batteries.

Comparing the required future capacity of wind and solar for different countries to their historical values, we see that many countries need a significant acceleration to reach a European-wide net-zero emissions target by 2050. This suggests that policymakers should place emphasis on avoiding potential barriers that slow down installation rates (e.g., inefficient regulatory frameworks or permitting processes, and misuse of public consultation processes). The requirements for accelerating growth are lessened under the self-sufficiency target for previously net-exporter countries and increased for net-importer countries. Ultimately, while the final transition path chosen by each country will depend on many social and political factors, the results presented here emphasize that the goals of one country could have significant implications for many others, necessitating a common planning strategy for the European energy system. Each country should also develop plans not only for renewable technologies development, but synthetic fuel production, mainly hydrogen, oil, and methanol, as they are the main energy carriers that will be traded in the future.

### Methods

#### Modeling approach and scenario design

We employ PyPSA-Eur, an open-source model, to simulate the sector-coupled European energy system. This model utilizes various datasets to represent demand across different sectors in Europe. It then constructs an energy system consisting of diverse generation technologies to meet this demand efficiently. The following focuses on describing the overall structure and core capabilities of the model, along with the additional features developed specifically for this study. Detailed

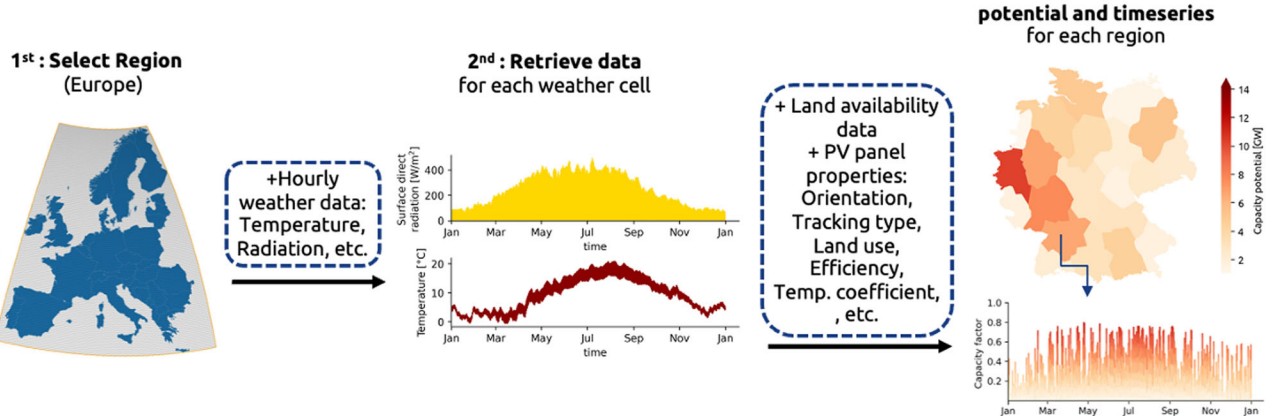

**Fig. 7 | Data-driven modeling for renewable resources.** Representation of the atlite package process for calculating the solar potential and the solar generation time series for different regions in Germany[61].

information regarding the structure and datasets used in the model is available in previous publications[35,60].

PyPSA-Eur utilizes linear equations and constraints, resulting in a linear and therefore convex optimization problem. Two different modeling approaches are employed in this paper. Firstly, an overnight greenfield optimization imposing a 95% $CO_2$ emissions reduction is used to identify which alternative PV configurations are more cost-effective. These are used in the second approach. Second, a myopic optimization is used to model the system from 2025 to 2050 with steps every 5 years. The initial year includes the existing capacities of solar, wind, and nuclear in Europe. The $CO_2$ limit imposed in every planning horizon is determined based on a carbon budget for Europe corresponding to a 1.7 °C temperature increase and assuming exponential decay of $CO_2$ emissions (see Supplementary Fig. 3). All modeled scenarios are summarized in Table 1.

The optimization problem includes various constraints, which comprise limiting $CO_2$ emissions, limiting transmission expansion, and limiting $CO_2$ sequestration in underground stores, consisting mostly of salt caverns[60]. Equation (1) represents the energy balance constraint, ensuring equilibrium between demand and generation at every node $i$ and time step $t$.

$$\sum_r g_{i,r,t} + \sum_s (h_{i,s,t}^- - h_{i,s,t}^+) + \sum_k \eta_{i,k,t} f_{k,t}$$
$$+ \sum_\ell K_{i\ell} f_{\ell,t} + \sum_p m_{p,t} = d_{i,t} \quad \leftrightarrow \quad \lambda_{i,t} \quad \forall i, t, \quad (1)$$

where $g_{i,r,t}$ is generator dispatch of technology $r$ at time $t$ and location $i$, and $h_{i,s,t}^-$ and $h_{i,s,t}^+$ are the discharge and charge of storage unit $s$, respectively. $f_{k,t}$ is dispatch of energy converter technology $k$, such as heat pumps converting electricity to heat, and $\eta_{i,k,t}$ is the efficiency of the technology $k$ to represent conversion losses. $K_{i\ell}$ is the incidence matrix of the energy transmission networks, such as AC and DC transmission lines or hydrogen pipelines, which has non-zero values equal to $-\eta_{i\ell}/\eta_{i\ell}$ when line $\ell$ is importing/exporting energy to or from node $i$, where $\eta_{i,\ell}$ is the efficiency of the pipe or transmission line, and $f_{\ell,t}$ is the imported/exported energy. $m_{p,t}$ is the equivalent energy of fuel $p$ such as methanol and oil that is imported to, or exported from, location $i$. $m_{p,t}$ is used to model energy carriers whose trade is assumed to be unlimited. $d_{i,t}$ is demand from electricity, heating, transport, industry, and agriculture at location $i$. $\lambda_{i,t}$ is the Lagrange multiplier of the constraint, which can be interpreted as the price of the respective energy carrier at location $i$ at time $t$[60].

### Self-sufficiency constraint
Following the approach proposed by van Greevenbroek et al.[13], we implement the equity constraint, which requires that each country generates a share $c_{equity}$ of its own demand annually, as shown in Equation (2).

$$\sum_{r,t} g_{i,r,t} + \sum_{s,t} (h_{i,s,t}^- - h_{i,s,t}^+) +$$
$$\sum_{k,t} \eta_{i,k,t} \cdot f_{k,t} \geq c_{equity} \cdot \sum_t d_{i,t} \quad \forall i \in \text{nodes}_{country} \quad (2)$$

The constraint is individually applied to each country by totaling production and demand across all sectors for the entire year. As different energy forms (electricity, thermal energy, hydrogen, etc.) are not segregated, a country might generate electricity to offset its oil demand, which is discussed in the results. However, implementing Equation (2) as a constraint is complex due to various system losses, mainly from cyclic efficiencies of batteries, hydrogen stores, and water tanks. Equation (3) modifies the constraint's formulation from Equation (2), by substituting demand from Equation (1), to instead limit each country's net import relative to its total energy production, reducing complexity and computational burden.

$$\left[\sum_{r,t} g_{i,r,t} + \sum_{s,t} (h_{i,s,t}^- - h_{i,s,t}^+) + \sum_{k,t} \eta_{i,k,t} \cdot f_{k,t}\right] \cdot (1 - 1/c_{equity}) +$$
$$\left[\sum_{\ell,t} K_{i\ell} f_{\ell,t} + \sum_{p,t} m_{p,t}\right] \leq 0 \quad \forall i \in \text{nodes}_{country} \quad (3)$$

Where the first bracket is the same as the left side of Equation (2) and represents energy production at node $i$. The second bracket is the sum of net energy imports from gas, hydrogen, and electricity networks, plus the net imports of fuels such as oil and methanol, which are either used directly by industry or by technology $k$ for conversion of energy. It is noteworthy that as the constraint is implemented on an annual basis, there are still time periods when a country relies entirely on energy imports.

### Modeling alternative solar PV configurations
We use the atlite package[61] (Fig. 7) to transform weather data into time series for onshore wind, offshore wind, and solar PV, as well as to

**Table 1 | Summary of scenario assumptions**

| Assumption | Solar competition | Transition to self-sufficiency |
|---|---|---|
| Optimization | Overnight (greenfield) | Myopic (brownfield) |
| Spatial resolution | 37 nodes (370 regions for renewable) | |
| Temporal resolution | 2-hourly | |
| Technology costs | 2050 | Investment year |
| $CO_2$ target | 95% reduction of emissions compared to 1990 level | 1.7 °C carbon budget with a net-zero goal by 2050 |
| Weather year | 2013 | |
| Transmission expansion | Today's capacity expandable by 10% | |
| Sectors | Electricity+heating+industry+agriculture+shipping, aviation, and land transport | |
| PV potential | 20 TW for utility PV and 1.5 TW for distributed PV | |
| Solar configurations | With all alternative solar configurations | With default/selected solar configurations |
| Self-sufficiency target | 0%/100% | 0%/100% target by 2050 (starting from 40% in 2025) |

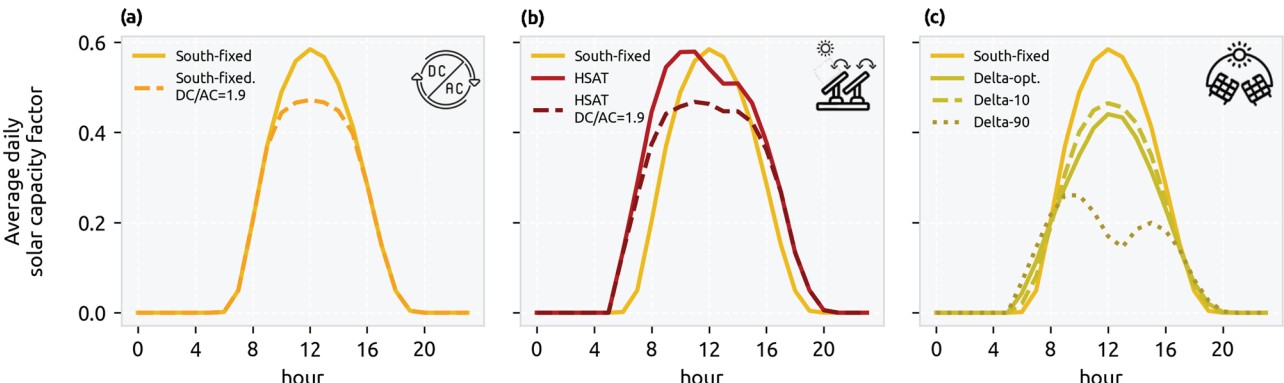

**Fig. 8 | Generation profiles for alternative solar PV configurations.** Hourly capacity factor for a day (average throughout the year) for a sample node in Spain for **a** fixed PV modules facing south and country-wise optimal inclination with 1 and 1.9 DC/AC ratio (ratio of installed solar DC capacity to AC power rating of the inverter), **b** horizontal single-axis tracking with 1 and 1.9 DC/AC ratio, and **c** delta configuration with 10°/optimal/90° inclination. Note that optimal inclination refers to the country-wise optimized inclinations for south-facing PV modules.

**Table 2 | Alternative solar PV technologies**

| Configuration | Investment (€/kW$_{AC}$) (2030) | O&M (% of investment) | LCOE[a] (€/kWh$_{AC}$) (2030) | Land use[b] ($MW_{DC}km^{-2}$) | Solar plant type |
|---|---|---|---|---|---|
| South-oriented (DC/AC = 1) | 320.9[64] | 2.47%[64] | 0.0295 | 102[64] | Utility/Distributed PV |
| South-oriented with inverter dimensioning (DC/AC = 1.25) | 383.7[64] | 2.47%[64] | 0.0283 | 81.6[64] | Utility/Distributed PV |
| HSAT (DC/AC = 1) | 377.5[64] | 2.28%[64] | 0.0274 | 88.8[68] | Utility PV |
| HSAT with inverter dimensioning (DC/AC = 1.25) | 454.5[64] | 2.28%[64] | 0.0270 | 71[68] | Utility PV |
| Delta configuration | 250.3[c] | 2.47%[64] | 0.0261 | 138.9[c] | Utility/Distributed PV |

Lifetime of 40 years is assumed for PV modules and 10 years for inverters.
Discount rate of 7% is assumed for utility PV plants and 4% for rooftop PV installations (see Supplementary Table 1 for cost assumptions of all years).
[a]To calculate the LCOE, an average location with annual global horizontal irradiation of 1085 (kWhm$^{-2}$) is considered.
[b]The estimates made here regarding land use could vary greatly for different PV plants based on location, PV modules efficiency, etc.[68,69]
[c]Based on own calculations (see Supplementary Note 3).

estimate the potential capacity that can be installed in every region. The latter is estimated based on the available area, considering suitable types of land (see Note S3.2 in Victoria et al.[35] for selected categories in Corine Land Cover database) and discounting the Natura 2000 protected areas. For onshore wind, 30% of the available area is considered to estimate the potential and land use of 10 MW per square km is assumed. Wind velocity at 100 m is read from ERA5 reanalysis data[62] and scaled to hub height of the selected wind turbines (Vestas V112 3MW for onshore wind and NREL 5MW for offshore wind). The power curve of turbines is then used to calculate wind power generation[60,63].

For solar PV, 10% of the available area is considered to estimate the potential and land use of 102 MW per square km[64] is assumed. Solar PV capacity factors are determined using satellite-aided SARAH-2 data for irradiance and temperature and assuming a temperature-dependent module efficiency equal to 20% at 25 °C[65]. For the default south-oriented PV configuration, the optimal inclination angle is selected for every country to ensure fair competition between fixed south-facing modules and other PV configurations. Analysis shows the optimal configuration with the highest generation for all countries is south-facing, with a tilt between 30° and 35° (see Supplementary Fig. 6).

In our model, solar installations comprise utility PV power plants connected to high-voltage (HV) buses and distributed PV plants, also known as rooftop PV, connected to low-voltage (LV) electricity buses. Each region has an HV and LV bus connected by a link representing the distribution grid[19]. On top of the south-oriented with optimal inclination, we analyze several alternative PV configurations. First, we consider inverter capacities lower than PV module capacity, known as inverter sizing. Second, we model horizontal single-axis tracking (HSAT) for utility PV plants. Third, we add delta configuration (with 10° optimal and 90° tilt) for both utility and distributed PV. Fourth, we introduce east, west, southeast, and southwest-facing modules for rooftop PV. The daily generation profiles in Spain for a south-oriented configuration, HSAT, delta, and a south-oriented configuration with a DC/AC ratio of 1.9 are compared in Fig. 8. Assumptions for these configurations are summarized in Table 2.

## Data availability
The raw and processed data used in this study, including network files for all main scenarios plus summarized tables containing key results, have been deposited in the Zenodo repository 14620646 under a CC-BY-4.0 license. Source data are provided with this paper.

## Code availability
The model is implemented by the open energy modeling framework PyPSA and makes use of the model PyPSA-Eur v0.9.0 (available under MIT license via github.com/PyPSA/pypsa-eur[66]) and the costs and technology assumptions included in the technology-data v0.8.1 (github.com/PyPSA/technology-data[67]). The source code and scripts to reproduce the results and figures included in this paper are publicly available at: https://github.com/Parisra/Solar-Transition-Paper.

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

## Acknowledgements

P.R. and M.V. acknowledge partial funding by the AURORA project, supported by the European Union's Horizon 2020 research and innovation programme under grant agreement No. 101036418. Figures 2 and 8 of this study include icons designed by Paul J. (DC to AC) and smashicons (other icons) from flaticon.com.

## Author contributions

P.R. designed the analysis, drafted the manuscript, and contributed to the analysis and interpretation of data. M.V. and E.Z. contributed to the modeling and provided substantial reviews of the manuscript. M.V. contributed to the conceptualization of the study and supervised the investigation.

## Competing interests

The authors declare no competing interests.
