## [Transparent Peer Review file · Nature Communications]

Strategic deployment of solar photovoltaics for achieving self-sufficiency in Europe throughout the energy transition

Corresponding Author: Ms Parisa Rahdan

Version 0:

Reviewer comments:

Reviewer #1

(Remarks to the Author)

The topic is interesting with new and significant perspectives on possible transnational energy system developments. Especially the modeling of new PV configurations and their role in a macro-energy system model look noteworthy to be published. The results are presented in a sophisticated way with nice and meaningful graphs. The optimization framework and the model parameters are openly available to reproduce the study, which is a clear advantage for the transparency of this work.

As the study is based on a comprehensive model it becomes especially important to elaborate and discuss the most important assumptions that can influence the results and the policy recommendations. This does not mean that they should not be taken, or the model itself must be necessarily revised. Nevertheless, the article and its results could be designed to be more transparent for the reader. Moreover, the logical structure of the paper could be enhanced at some points. This has been addressed in my review comments below.

Comment 1:

There is a lack of a more comprehensive discussion section in this work, where not only results are discussed but also basic assumptions and their expected influence on the results are evaluated. For instance, could you please address the following points in more detail:

- According to previous studies (e.g. <https://doi.org/10.1016/j.energy.2014.02.109>) the transnational electricity exchange is especially beneficial to the European system due to the smoothing effect of geographical dispersion on wind power fluctuations. Fully renewable-based energy systems require high-capacity expansion of the transmission grid. The upper bound on 10% transmission grid expansion might decrease the benefit from wind expansion (In comparison, other European energy system studies such as <https://www.sciencedirect.com/science/article/pii/S2542435123002660> or <https://www.sciencedirect.com/science/article/abs/pii/S0301421523002318> have much higher grid expansion). In comparison, solar simultaneity across countries is high and can hence be less exploited to exchange electricity via the grids, but rather being stored or converted to hydrogen or synfuels (compared to wind). Together, this could lead to model that favors PV expansion.
- As wind is the renewable energy resource with higher variability and heterogeneity across European countries, the overall system can also benefit substantially in terms of costs if the right technology (hub height, rotor diameter, etc.) is chosen at the right location resulting in very heterogenous profiles. By making available only one wind turbine technology, the system benefits from wind might be underestimated.
- Also, the capacity ratio (approx. 4:1) between solar and wind in your results might be related to these two points. Can you compare this ratio to other European Energy system models for 2050?

The cost increase due to sufficiency constraints is benchmarked against this sub-optimal European base model (with less customized wind technology selections and a bounded transnational electricity grid). As the difference could get a lot larger with adequate modeling of the European wind system with high transnational electricity transmission, and this would have major implications on the interpretation of your results this should be discussed in more detail.

Comment 2:

Three research questions have been selected to be analyzed with the developed model. This is fine, but the logical transition between research gaps, the research questions (1.), and their results (2.1 – 2.3) can be enhanced to be clearer. How do the questions and sections relate to each other? Sometimes also simple mind bridges can help to get a more coherent storytelling.

In my perspective, e.g. for the results this could be easier by changing the order of the individual results (2.2 2.3 2.1)

Comment 3:

Is there evidence from other studies that the value of ("only") 10% of the available area for utility PV is really a low value? Are there conflicts of interests for these free areas?

Comment 4:

Table 2 is nice showing the parameters of your technologies but does not underline your message regarding the value of alternative PV configurations. It might help to additionally compare the LCOEs of different technologies to highlight their value.

Comment 5 - Structural points:

- To place the Method Section as 4th chapter does not make sense to me. Is this somehow required?
- After mentioning the three main novelties of the study, the summary of the results at the end of the introduction should be removed. It would be preferable to focus on elaborating the novelties and on describing the structure of the chapters.

Comment 6 - Minor points:

- The scripts to reproduce the results could not be found under the indicated link (<https://github.com/Parisra/solar-transition-paper>)
- You could adjust the text description of equation 1 to the order of the elements in the formula (easier to read)
- Typo Table 1: "Mopic"

Reviewer #2

(Remarks to the Author)

The work focuses on the highly relevant issue of energy security and the energy transition in Europe, presenting an innovative approach to addressing the problem. The analysis emphasizes the feasibility of accelerating renewable energy deployment rates based on recent trends, despite existing challenges. This study makes a valuable contribution to energy system modeling by integrating emerging solar configurations and incorporating self-sufficiency constraints. It provides practical insights into balancing regional equity, cost-efficiency, and energy security.

Compared to established literature, this study extends previous analyses by incorporating more granular modeling (e.g., inverter dimensioning, regional PV configurations) and employing a dynamic approach to transitioning towards self-sufficiency. The conclusions are well-supported by robust data and modeling. The integration of historical trends with scenario analyses provides a solid foundation for the claims.

Suggestions for improvement:

1. Detailed datasets: The datasets for specific assumptions, such as cost reductions, could be elaborated in greater detail to ensure clarity and reproducibility.
2. Citation format: Citations should not be grouped together; instead, they should be presented individually to provide more precise references.
3. Equation S2: The cost parameter appears to be missing. Including its value is essential for a complete understanding of the equation.
4. Figures S12 to S21: The small chart sections in these figures are not clearly visible due to the color or scale. Adjusting the scale or improving the color contrast would enhance readability.
5. Methodology rigor: The methodology is rigorous, utilizing the PyPSA-Eur model to analyze sector-coupled systems at high spatiotemporal resolution. The inclusion of innovative solar PV configurations is a particularly commendable enhancement.
6. Empirical validation: Additional empirical validation, especially for novel configurations such as delta PV, would strengthen the study's findings and provide further credibility to the conclusions.

These improvements would significantly enhance the clarity, precision, and impact of the paper.

Version 1:

Reviewer comments:

Reviewer #1

(Remarks to the Author)

Thank you for your thorough revisions and detailed elaborations. Your sensitivity analyses in the supplementary information have effectively addressed my concerns, demonstrating a high level of transparency and a robust methodological approach. Regarding "Comment 3" from my initial review, and based on your responses where you state that these limits are rather arbitrary: I believe the removal of the evaluating word "only" in line 395 could have been considered. However, this is a minor point, as your assumptions are already well contextualized in line 144. Overall, all comments have been addressed in an exemplary manner and relevant information has been added to the manuscript.

In summary, this study makes a valuable contribution to the energy system modeling research community by integrating emerging solar configurations and exploring important aspects of system analysis, such as self-sufficiency constraints in a carbon-neutral Europe.

I am pleased to recommend this study for publication. Congratulations!

Reviewer #2

(Remarks to the Author)

Thank you for considering the revisions. After reviewing the changes made by the author, I find the paper suitable for publication.

Response to Reviewer #1:

Comment 1:

There is a lack of a more comprehensive discussion section in this work, where not only results are discussed but also basic assumptions and their expected influence on the results are evaluated. For instance, could you please address the following points in more detail:

Thank you very much for taking the time to review our paper. Your suggestions were a great help to us in revising and improving the paper. We detail below our responses along with the revisions that were carried out based on each comment.

- According to previous studies (e.g. <https://doi.org/10.1016/j.energy.2014.02.109>) the transnational electricity exchange is especially beneficial to the European system due to the smoothing effect of geographical dispersion on wind power fluctuations. Fully renewable-based energy systems require high-capacity expansion of the transmission grid. The upper bound on 10% transmission grid expansion might decrease the benefit from wind expansion (In comparison, other European energy system studies such

as <https://www.sciencedirect.com/science/article/pii/S2542435123002660> or <https://www.sciencedirect.com/science/article/abs/pii/S0301421523002318> have much higher grid expansion). In comparison, solar simultaneity across countries is high and can hence be less exploited to exchange electricity via the grids, but rather being stored or converted to hydrogen or synfuels (compared to wind). Together, this could lead to model that favors PV expansion.

- As wind is the renewable energy resource with higher variability and heterogeneity across European countries, the overall system can also benefit substantially in terms of costs if the right technology (hub height, rotor diameter, etc.) is chosen at the right location resulting in very heterogenous profiles. By making available only one wind turbine technology, the system benefits from wind might be underestimated.

- Also, the capacity ratio (approx. 4:1) between solar and wind in your results might be related to these two points. Can you compare this ratio to other European Energy system models for 2050?

The cost increase due to sufficiency constraints is benchmarked against this sub-optimal European base model (with less customized wind technology selections and a bounded transnational electricity grid). As the difference could get a lot larger with adequate modeling of the European wind system with high transnational electricity transmission, and this would have major implications on the interpretation of your results this should be discussed in more detail.

Response: As you have pointed out, the solar to wind ratio is an important aspect when it comes to results, so we will address this subject fully both here and in the main text to clarify the matter for readers:

1. The main reason behind the increase in solar capacity/generation in this study compared to previous research that used PyPSA-Eur is the **reduction in the costs of solar PV**. The Danish Energy Agency (DEA) [1], which is used as a source for our cost assumptions, states that the investment cost for utility PV is 380.4 €/KW in 2030. However, this assumption is for a module with a DC/AC ratio of 1.25. This was previously not considered in the model and was overlooked. A DC/AC ratio of 1.25 means having 1.25 kW of solar panels paired with a 1 KW inverter, the potential generation for which is on average 24.6% higher than a 1 KW panel paired with the same inverter, as shown in Supplementary Figure 4:

Supplementary Fig.4 Duration curve of capacity factors for a) south-facing fixed solar PV with DC/AC ratio of 1 and 1.5 (both DC and AC generation are shown),

Therefore, a panel with 380 €/kW cost needs to be paired with 24.6% higher generation in the model. In other words, if we now calculate how much a 1 KW panel paired with 1 KW inverter would cost according to the DEA (as detailed in Supplementary equation 2) :

$$\text{DC component costs (Module + installation + soft costs + other) + AC component costs (Inverter + Transformer and grid connection) = 251.3 + 69.5 = 320.8 \text{ €/kW_AC}}$$

This means that in our scenarios, we have effectively decreased the investment cost for utility solar **from 380.4 €/kW_AC to 320.8 €/kW_AC**. As a result of the 16% cost decrease the system's solar to wind ratio increases. Also, currently solar PV has an investment cost decline stronger than many studies have assumed in the past. We have now added a section to the main text with a more detailed explanation, reproduced here:

Including inverter dimensioning effectively reduces the investment cost of utility solar PV in our model by 15.8% relative to a PV system with the same DC and AC capacity. This decrease in costs causes the system to have a 1.7 ratio for solar generation to wind generation, which is more than twice what was observed in previous similar studies [2]. While this ratio seems high, recent studies incorporating recent drops in solar PV costs have shown even higher ratios, up to 1.9 [9]. However, as this ratio changes with different cost assumptions, we will focus on robust results in this section, such as the role of methanol and hydrogen. Refer to Supplementary Notes 3 and 4 for more details.

A more detailed explanation was also added to the Supplementary:

Supplementary Note 3: Cost and land-use calculations for alternative solar configurations

1. Inverter dimensioning

In a PV installation, one part of the capital cost depends on the DC capacity (e.g. PV modules, land-use costs, soft costs), represented in Supplementary Eq. (2) as DC components capital cost $t_{\frac{DC}{AC}=1}$, while the other part depends on the AC capacity of the

plant (e.g. inverter, grid-connection), represented as

AC components capital cost $\frac{DC}{AC}=1 \left(\frac{\text{€}}{\text{MW}_{AC}} \right)$. Therefore, for a PV power plant with a DC/AC ratio of r :

$$\begin{aligned} \text{PV power plant capital cost}_{\frac{DC}{AC}=r} \left(\frac{\text{€}}{\text{MW}_{AC}} \right) = \\ \text{DC components capital cost}_{\frac{DC}{AC}=1} \left(\frac{\text{€}}{\text{MW}_{DC}} \right) \cdot r \left(\frac{\text{MW}_{DC}}{\text{MW}_{AC}} \right) + \\ \text{AC components capital cost}_{\frac{DC}{AC}=1} \left(\frac{\text{€}}{\text{MW}_{AC}} \right) \end{aligned}$$

To see the impact of the new calculation method, which incorporates inverter sizing, consider the following example: In previous studies using PyPSA-Eur, the investment cost for utility-scale PV in 2030 was assumed to be 380.4 €/kW, based on DEA [1]. However, this cost corresponds to a module with a DC/AC ratio of 1.25. This detail can be easily overlooked, as pairing a 1 kW solar panel with a 0.8 kW inverter results in less than a 3% loss in annual solar generation. By applying Equation (2), the investment cost for utility-scale solar PV is recalculated to 320.8€/kW_{AC} (251.3€ for the DC components + €69.5€ for the AC components), effectively reducing the investment cost by 15.8% relative to a PV system with the same DC and AC capacity.

Detailed assumptions for DC and AC components' capital cost for different investment years are shown in Supplementary Table 1. The final capital cost for fixed panels and horizontal single-axis tracking (HSAT) with different inverter ratios as calculated by Supplementary Eq. (2) is also shown in Supplementary Table 1.

2. Based on your excellent points regarding transmission expansion and the inclusion of other wind turbines, we have now included two sensitivity analyses to show the effect of these parameters on the system, which we will discuss further. Before that, we would like to clarify that the 10% expansion limit was chosen in order to represent the current slow build-up of transmission lines and grid expansion acceptance issues across Europe. This fact is also mentioned by Gawlick et al. [3]:

“Scenario expensive electricity transmission investigates the impact of investment hurdles for electricity transmission projects by increasing the investment costs due to longer planning periods and possibly re-planning measures. For example, expanding the national grid in Germany by a new DC connection between North and South Germany roughly three-folded the costs after the originally planned overhead line had to be transformed into an underground cable due to several environmental and social issues (Buergerdialog Stromnetz, 2019).”

The figure below shows the results of the sensitivity analysis for **higher transmission expansion** allowance for overnight runs representing the 4 main scenarios. The analysis was done only for overnight due to the long computational time.

Figure 1- Comparison of energy generation of major renewable technologies for overnight runs representing the four main scenarios (Base, Base with 100% self-sufficiency target, Alternative solar configurations, and Alternative solar configurations with 100% self-sufficiency target) under different transmission expansion allowances.

As shown in the figure, with unlimited expansion and no new solar configurations (Base case), the solar to wind generation ratio decreases by only 4.3%. This decrease is lowest when including new solar configurations into the model without any self-sufficiency target. Therefore, this ratio can change slightly under different assumptions but the main results discussed in the paper, such as having more solar PV and wind under the self-sufficiency target or HSAT and inverter dimensioning being cost-efficient, remain robust for all scenarios.

As for different wind turbines, the ‘atlite’ package [4], which we use for modeling the potential generation of renewables, includes many different turbines with different power curves. For example, below is the normalized power curve of 4 different turbines:

Figure 2- Normalized power curves for 4 different onshore wind turbines.

However, the main issue we have with using different turbine types is cost estimation. A turbine that has lower capacity and cuts off power generation at lower wind velocity (e.g. Suzlon 1.5 MW turbine) than our default turbine (VESTAS 3 MW turbine), should have a lower cost in order to be selected by the system for an area where wind speeds are lower. It was difficult to find good cost estimates for the turbine Suzion 1.5MW so we decided to conduct another sensitivity analysis. In the figure below, we have included the ‘NREL 4-MW’ wind turbine [5], which shows higher generation for low wind speeds than all other turbines, so much so that it can increase potential generation (annual capacity factor) in certain areas up to 30%. According to NREL ATB 2020 [5], this turbine is estimated to cost 862, 1173, and 1338 \$/kW for the advanced, moderate, and conservative scenarios, respectively. Using the same costs for other parts of the wind powerplant as DEA (Danish Energy Agency) that are used for the default turbine, the NREL 4-MW would be 16-58% more expensive than the VESTAS 3 MW. Therefore, we added this new turbine with two different cost estimates, one equal to our default onshore wind turbine, and one one 44% more expensive (equal to the moderate scenario in NREL ATB 2020). All scenarios are modeled with unlimited transmission expansion allowance.

For the "low-cost" scenario, the NREL-4MW becomes the dominant wind technology across all areas, shifting the system toward increased onshore wind capacity. As a result, the solar-to-wind ratio decreases by 51% in the Base scenario. However, when the NREL-4MW turbine's costs are 44% higher, the outcomes closely resemble those of the main scenarios, with a maximum reduction of 6% in the solar-to-wind ratio compared to the scenarios in Figure 1 with unlimited expansion. Therefore, the energy mix could undergo substantial changes if a higher-yielding or more cost-effective onshore wind turbine becomes commercially available.

Figure 3- Comparison of energy generation of major renewable technologies for overnight runs representing the four main scenarios (Base, Base with 100% self-sufficiency target, Alternative solar

Finally, we have included in the supplementary Note 5 a summary of these two sensitivity analyses along with a comparison between the solar to wind ratio in our study to similar studies. This section aims to clarify that future studies with better representations of wind turbines could easily see different results in terms of the generation mix, but the main results discussed in the paper remain valid under different assumptions:

Supplementary Table 2- Comparison between the ratio of total installed solar capacity (utility and rooftop) to total installed wind capacity (onshore wind and offshore wind) in the Base overnight scenario of this study to recent similar studies

Study	Main assumptions	Solar to wind ratio (Installed capacity)
This study (Base overnight scenario)	Sector-coupled, 95% emissions reduction, 2050 cost assumptions, 10% grid expansion	1.84
Gotske et. al (2024) [6]	Sector-coupled, net-zero target, 2030 cost assumptions, average of 60 weather years, no grid expansion (see Supp. Figure 39)	1.9
Neumann et. al (2023) [7]	Sector-coupled, net-zero target, 2030/2050 cost assumptions*, zero/unlimited grid expansion	2030 costs: 1.6 (0)/1.42 (unlim.) 2050 costs: 2.7(0) /2.2 (unlim.)
Gawlick & Hamacher (2023) [3]	Electricity and hydrogen sectors only, net-zero target, 2050 cost assumptions from IEA**, unlimited grid expansion (see Table A.9, ‘totally flexible’ scenario)	1.6
* 2050 assumptions include cost reductions of solar photovoltaics by 25% and onshore wind by 7% relative to 2030 values		
** Onshore wind to solar cost is 3.5 based on this data, 3.4 based on DEA without consideration of inverter dimensioning, and 4.1 when reducing solar cost by considering inverter dimensioning		

Supplementary Table 3- Comparison between the ratio of solar (utility and rooftop) to wind (onshore wind and offshore) in terms of installed capacity and energy generation for the Base myopic scenario of this study to recent similar studies

Study	Main assumptions	Solar to wind ratio (Energy generation)	Solar to wind ratio (Installed capacity)
Base Myopic scenario	2050 with net-zero target under 1.7° temperature increase carbon budget starting from 2025	1.69	3.98
Zeyen et. al (2023) [2]	2050 with net-zero target under 1.7° temp. increase budget from 2020 (Fig. S31)	0.65 (ranging from 0.39 to 0.77 with different learning rates for solar, wind, and electrolysis)	1.36 (ranging from 0.85 to 1.69)
Bogdanov et. al (2019) [8]	2050 with net-zero target, global scenario (Supp. Table 12)	1.48	3.5
Breyer et. al (2023) [9]	2050 with net-zero target, sector-coupled for Europe, scenarios: moderate /leading (Table 1)	1.88 / 1.92	4.7 / 4.5

We have also added a paragraph at the end of the results section to address the different limitations of this study that could be improved in future research:

2.4. Limitations and further work

Some limitations of this study, such as limiting transmission expansion to 10% of current volumes or representing wind generation using only one onshore and one offshore turbine types, are discussed in Supplementary Note 5. Overall, while varying assumptions may alter the system's structure in terms of installed capacity and generation, the core findings regarding the cost-efficiency of new solar configurations and the changes needed to achieve self-sufficiency remain robust. Future research could explore the impact of extreme weather events on self-sufficiency requirements for individual countries, or assess the feasibility of imposing annual self-sufficiency constraint for a shorter time period (e.g., weekly or daily).

Comment 2:

Three research questions have been selected to be analyzed with the developed model. This is fine, but the logical transition between research gaps, the research questions (1.), and their results (2.1 – 2.3) can be enhanced to be clearer. How do the questions and sections relate to each other? Sometimes also simple mind bridges can help to get a more coherent storytelling. In my perspective, e.g. for the results this could be easier by changing the order of the individual results (2.2 \diamond 2.3 \diamond 2.1)

Response: Many thanks for this great insight. We had some debate about this topic when organising the results section, and the rationale for choosing the current sequence, rather than the more conventional order (2.2-2.3-2.1), was to present the following storyline:

1. **Establish the impact of self-sufficiency on the system capacity mix:** Highlight the importance of solar PV in achieving self-sufficiency target, leading to exploring new solar configurations
2. **Introduce cost-efficient solar configurations:** Integrate new solar configurations into the model, and revisit the self-sufficiency experiment with these configurations included.
3. **Conclude with a discussion of the overall transition path:** Analyze the transition trajectories for each country across all scenarios, discuss how new solar PV configurations can facilitate this transition, and whether achieving the targets is feasible based on historical growth rates.

We believe this to be a more coherent narrative, linking the topics of self-sufficiency and innovative solar configurations. We have now made modifications to the text that hopefully

improve this storyline and clarify the reasoning for the readers. For the sake of clarity, we reproduce here in red the additional text that we have added to the paper:

2. Results and discussion

This section is organized as follows: First, we examine the impact of implementing self-sufficiency on the energy system at European and country-specific levels. After establishing the critical role of solar PV in achieving self-sufficiency, we proceed to the second part, where we use an overnight scenario to evaluate various new solar configurations. This analysis identifies the cost-efficient configurations that should be incorporated into the model. Third, we revisit the self-sufficiency experiment using these newly selected configurations to assess their impact on the system, both with and without the self-sufficiency constraint. Finally, we conclude by analyzing the transition pathways for wind and solar energy across different countries under various scenarios, evaluating whether the required ambitions are achievable based on historical growth rates.

...

Notably, solar PV is the only technology whose capacity consistently increases across all countries that need to boost local generation to achieve self-sufficiency (Fig. 1b). Given the significant deployment of both utility-scale and distributed solar capacities, we will explore the potential of alternative solar technologies to improve energy independence in Sections 2.2 and 2.3.

2.2. Solar PV tracking and inverter dimensioning can reduce costs for the green transition

As our model uses 370 regions to assess solar resources, adding each new solar configuration increases computational complexity. Therefore, we first perform two simplified, highly renewable overnight scenarios to determine which configurations merit further investigation under the transition scenario. In both scenarios, 19 alternative solar PV configurations are evaluated, which were chosen based on an initial cost-benefit analysis (Fig. 2).

...

This means that the system still does not require extra generation at noon, and is shifting the capacities previously installed in Spain to other countries. The capacity shifts that happen under self-sufficiency are quite significant in some countries. We will conclude the results in the next section by discussing whether these targets are realistically attainable based on past performance and deployment trends.

...

2.4. Large installation rates are within reach

Now we turn our attention to how the transition scenarios play out for each country. We compare the historical capacity deployments of solar PV and wind (both onshore and offshore) using data from IRENA, with what is needed from now till 2050.

Comment 3:

Is there evidence from other studies that the value of (“only”) 10% of the available area for utility PV is really a low value? Are there conflicts of interests for these free areas?

Response: We implement two primary land-use limits for renewable energy in our model: 10% of "available land" for solar and 30% for wind. The calculation of "available land" is based on the Corine Land Cover database, excluding unsuitable categories such as natural reserves and urban areas. Two key considerations regarding these limits are as follows:

1. These limits are somewhat arbitrary and serve to prevent the installed capacity in any single region from exceeding levels deemed realistic. This precaution is necessary because in a cost optimization, the model can be quite aggressive in concentrating renewable installations in regions with marginally lower costs, even if the cost differences compared to neighboring regions are very small. An example of this aggressive behavior is evident in the nodal land-use distribution for the "Base" transition scenario in 2050, as shown in Supplementary Fig. 11 where the optimal solution concentrates the solar capacity in the Southern nodes of every country.

Supplementary Fig. 14: Results for the overnight scenarios (95% carbon emissions reduction, greenfield assumptions, technology costs for 2050) where the cost-efficiency of 19 alternative solar configurations is examined.

2. Cumulatively, our results found that only 2.3% of available land across Europe is projected to be used for solar installations by 2050, while 3.7% is allocated for wind. We can compare our values with those reported by the European Joint Research Center (JRC) [10], who, using very high geographical granularity, estimates that the amount of suitable

land for wind and ground-mounted solar power production constitutes **3.4%** of the EU's total surface area:

‘Protected nature sites and biodiversity areas, forestry and water bodies are excluded as suitable sites. The use of agricultural land for energy production is subject to strict limitations. Furthermore, buffer zones around infrastructure and settlements are enforced to minimise disturbance and the local phenomenon known as ‘not in my backyard’ (NIMBY), a community’s opposition to the possible impacts of a new renewable energy project’

Based on this, the 3.4% value reached by JRC is under such strict agricultural, environmental, and biodiversity constraints. Our model projects a total land use of 6% in available areas, equivalent to approximately 4.5% of the total EU land area. This is close to the JRC’s conservative estimate. However, in our results, under the self-sufficiency constraint, some countries may face significant challenges in meeting the required land use. This is also in accordance with JRC [10] who states:

‘Member States display strong variations in the availability of suitable land for new renewable energy installations, ranging from 0.1 % to almost 9 %. Considering both solar and wind, larger shares of suitable land are found in Latvia, Romania, Estonia, Lithuania, Cyprus and Portugal (above 5 % of each country’s total area), whereas in Malta, Austria, Slovenia and Belgium these shares are residual (below 0.5 %). ‘

We have now added a short summary of this discussion to the main text to clarify these assumptions and limitations. For the sake of clarity, we reproduced there the added text in red:

Solar PV and wind energy become the cornerstone of the transformed energy system, with solar PV being crucial for achieving self-sufficiency. By 2050, 5.1 TW of solar and 1.3 TW of onshore and offshore wind capacity are installed across Europe (see Supplementary Fig. S9), taking up 57% and 36% of the electricity generation, respectively. This would require approximately 2.3% of available land for solar PV and 3.7% for wind, equivalent to about 4.5% of Europe’s total land area. This land use aligns with very conservative estimates of suitable land for renewable energy installations that account for environmental, agricultural, biodiversity, and social constraints [10]. However, achieving these targets may be significantly more challenging for certain countries.

Comment 4:

Table 2 is nice showing the parameters of your technologies but does not underline your message regarding the value of alternative PV configurations. It might help to additionally compare the LCOEs of different technologies to highlight their value.

Response: This is a great insight. We have now added average LOCE to the Table. Hopefully this better conveys the cost efficiency of new solar PV configurations such as HSAT, which can achieve lower LCOE compared to fixed solar plants [11]:

Table 2: Alternative solar PV technologies

Configuration	Investment (€/kW_AC ₂₀₃₀)	O&M (% of investment)	LCOE* (€/kWh_AC ₂₀₃₀)	Land use** (MW_AC/km ²)	Solar plant type
South-oriented (DC/AC=1)	320.9	2.47 %	0.0295	102	Utility/ Distributed PV
South-oriented with inverter dimensioning (DC/AC=1.25)	383.7	2.47 %	0.0283	81.6	Utility/ Distributed PV
HSAT	377.5	2.28%	0.0274	88.8	Utility PV
HSAT with inverter dimensioning (DC/AC=1.25)	454.5	2.28%	0.0270	71	Utility PV
Delta configuration (10° tilt)	250.3 [†]	2.47 %	0.0261	138.9 [†]	Utility/ Distributed PV

Lifetime of 40 years is assumed for PV modules and 10 years for inverters. Discount rate of 7% is assumed for utility PV plants and 4% for rooftop PV installations (see Supplementary Table S2 for cost assumptions of all years).

* To calculate the LCOE, an average location with 1,085 annual peak sun hours is considered.

** The estimates made here regarding land use could vary greatly for different PV plants based on location, PV modules efficiency, etc.

† Based on own calculations (see Supplementary section S3).

We also include here a simple comparison between the LCOE of wind and solar in our model for 2025 compared to current market values. The table below shows the LCOE of solar configuration, plus onshore wind and offshore wind achieved in the model for **2025**.

For comparison with current market values, global weighted average LCOE stood at 0.044 \$/kWh for utility-scale solar PV and 0.033 \$/kWh for onshore wind according to IRENA. Solar PV already achieved 0.034 \$/kWh and 0.036 \$/kWh in Australia and China in 2023 [12]. For European countries in 2023, these numbers stood at 0.060 \$/kWh (average of 7 markets) for utility solar PV, 0.046 \$/kWh (average of 15 markets) for onshore wind, and 0.067 \$/kWh for offshore wind, respectively [12].

Taking into account possible further reductions in cost from 2023 to 2025, the overall LCOE in our model for 2025 aligns well with the global average LCOE. The cost decline for solar is more pronounced than for wind when comparing our model's LCOE with today's global averages. This is because wind is highly location-dependent, whereas solar power is less sensitive to geographic variation. In our model, as more wind is installed in less optimal regions by 2025, the average capacity factor declines, increasing the LCOE. This can also be seen by comparing the capacity-weighted LCOE with the average LCOE. For solar, the difference is minimal since solar irradiation remains relatively consistent

across locations. For wind however, the capacity-weighted LCOE is significantly lower than the average LCOE, highlighting the impact of site selection.

Table – Comparison of model and market LCOE for solar and wind technologies

Source	Parameter	Solar (DC/AC =1)	Solar-HSAT (DC/AC= 1)	Onshore Wind	Offshore Wind (dc /ac connection)
This study	Capacity-weighted average LCOE ₂₀₂₅ (€/KWh)*	0.032	0.031	0.034	0.051/0.048
	Average LCOE ₂₀₂₅ (€/KWh)	0.035	0.033	0.061	0.055/0.067
	Average capacity factor (MWh/MW)	0.12	0.15	0.2	0.37/0.49
IRENA (2023)	Global average LCOE	0.044	-	0.033	
	Europe average LCOE (\$/kWh)	0.06	-	0.046	0.067
* average of four main scenarios					

The capital cost and FOM of the main technologies are also now added to the supplementary under the ‘Supplementary Tables’ section.

Comment 5 - Structural points:

- To place the Method Section as 4th chapter does not make sense to me. Is this somehow required?

Response: Yes, the Journal editorial mandate is to have the following section order: Title, Abstract, Introduction, Results, Discussion, Methods, etc.

We have now moved a small section of the methods describing the overall methodology to the beginning of the results section so that readers have a general understanding of the model and the optimisation before results are presented and discussed. We have also increased our references to methods throughout the text to give readers a reference for how the modelling was done. For the sake of clarity, the added text is reproduced in red below:

Results:

This section is organized in the following order...

We use an open-source model to optimize the capacity and dispatch of all system elements, aiming to minimize total system costs while adhering to defined constraints (refer to methods for a more detailed description). Covering the entire ENTSO-E area, we use a

spatial resolution of 37 nodes with 370 regions for renewable potential estimation and capture a one-year period with 2-hour time steps. The model integrates electricity generation including solar, onshore and offshore wind, hydropower, nuclear, methane gas, and storage technologies such as batteries, and transmission grid as well as a simplified distribution grid. Additionally, it incorporates the heating, land transport, aviation, shipping, industry (including industrial feedstock), and agriculture sectors, considering their specific demands and incorporating relevant technologies like heat pumps, electric vehicles, and industrial processes (see Supplementary Note 1 for more details).

Other small adjustments to refer to methods section more clearly:

*which requires that the self-sufficiency coefficient for every country (see Eq. 2 in **Methods**) reaches 60%, 80%, and 100% in 2030, 2040, and 2050,*

We then examine two overnight near-zero scenarios, both featuring the 19 configurations, but only one imposing a 100% self-sufficiency target. A summary of all the scenarios is available in Table 1.

For rooftop PV systems connected to the low-voltage grid, we consider the same configurations as utility except HSAT, and instead add southeast and southwest-facing modules with an inverter ratio of 1.9. Additional details regarding the costs, land use, and generation profiles of these new solar configurations are provided in the Methods section and Supplementary Notes 3 and 4.

We model again the transition paths including the alternative solar configuration that were found cost-effective in the previous overnight optimization exercise. These include south-oriented configuration with inverter ratios of 1.5 and 1.7 (both for utility-scale and distributed systems), HSAT with inverter ratios of 1.3 and 1.5, and delta configuration with 10° inclination and inverter ratio of 1.5. For a detailed explanation of the cost assumptions related to inverter dimensioning and delta configurations, please refer to Supplementary Note 3.

- After mentioning the three main novelties of the study, the summary of the results at the end of the introduction should be removed. It would be preferable to focus on elaborating the novelties and on describing the structure of the chapters.

Response: Thanks for letting us know this does not read well. Since a very short overview of results is typically done in this paper structure to further interest readers, we have kept this part but shortened it, and also added a description of sections to the end of the introduction. For the sake of clarity, we reproduce the added text below:

This work introduces three main novelties: ... The results indicate that self-sufficiency minimally impacts total costs but promotes more uniform capacity distribution. However, costs could rise by 150% for net-importer countries by 2050. High-value synthetic fuels are produced in renewable resource-rich countries, and hydrogen is traded extensively. Alternative PV configurations also aid self-sufficiency by reducing costs and extending generation. Ultimately, growth rates for both wind and solar require higher ambitions in many European countries, but the trends of recent years show that they are achievable.

The upcoming section will delve into the main results, beginning by discussing a baseline transition-to-net-zero scenario, followed by describing the impacts of introducing a self-sufficiency constraint and new PV configurations. Based on the results, we will make recommendations to energy modelers and policymakers in the conclusion section. Lastly, the methods section outlines the mathematical formulation of the self-sufficiency constraint and the key modeling assumptions, in particular regarding solar PV costs and alternative configurations.

Comment 6 - Minor points:

- The scripts to reproduce the results could not be found under the indicated link (<https://github.com/Parisra/solar-transition-paper>)

Response: All the raw data used in the study, plus the scripts used to generate the networks and the visualizations are now available at Zenodo and Github:

Data availability

Network files for all main scenarios plus summarized tables containing key network results are available at: [Zenodo repository 14620646](https://zenodo.org/record/14620646)

Code availability

The model is implemented by the open energy modeling framework PyPSA and makes use of the model PyPSA-Eur v0.9.0 and the costs and technology assumptions included in the technology-data v0.4.0. Scripts to reproduce the results and figures included in this paper are publicly available at: <https://github.com/Parisra/Solar-Transition-Paper>

- You could adjust the text description of equation 1 to the order of the elements in the formula (easier to read)

Response: This is a great point. We have adjusted the text to match the order in the equation :

Eq. (1) represents the energy balance constraint, ensuring equilibrium between demand and generation at every node i and time step t .

$$\sum_r g_{i,r,t} + \sum_s (h_{i,s,t}^- - h_{i,s,t}^+) + \sum_k \eta_{i,k,t} f_{k,t} + \sum_l K_{i\ell} f_{\ell,t} + \sum_p m_{p,t} = d_{i,t} \quad \Leftrightarrow \quad \lambda_{i,t} \quad \forall i, t$$

where $g_{i,r,t}$ is generator dispatch of technology r at time t and location i , and $h_{i,s,t}^-$ and $h_{i,s,t}^+$ are the discharge and charge of storage unit s , respectively. $f_{k,t}$ is dispatch of energy converter technology k , such as heat pumps converting electricity to heat, and $\eta_{i,k,t}$ is the efficiency of the technology k to represent conversion losses. $K_{i\ell}$ is the incidence matrix of the energy transmission networks, such as AC and DC transmission lines or hydrogen pipelines, which has non-zero values equal to $-\eta_{i,\ell}/\eta_{i,\ell}$ when line ℓ is importing/exporting energy to or from node i , where $\eta_{i,\ell}$ is the efficiency of the pipe or transmission line, and $f_{\ell,t}$ is the imported/exported energy. $m_{p,t}$ is the equivalent energy of fuel p such as methanol and oil that is imported to, or exported from, location i . $m_{p,t}$ is used to model energy carriers whose trade is assumed to be unlimited. $d_{i,t}$ is demand from electricity, heating, transport, industry, and agriculture at location i . $\lambda_{i,t}$ is the Lagrange multiplier of the constraint, which can be interpreted as the price of the respective energy carrier at location i at time t .

- Typo Table 1: “Mopic”

Response: Thanks. The spelling was corrected.

References

1. “Technology Data for Generation of Electricity and District Heating,” Danish Energy Agency, Tech. Rep., 2024. [Online]. Available: <https://ens.dk/en/our-services/projections-and-models/technology-data/technology-data-generation-electricity-and>

2. E. Zeyen, M. Victoria, and T. Brown, “Endogenous learning for green hydrogen in a sector-coupled energy model for Europe,” *Nature communications*, vol. 14, no. 1, p. 3743, 2023.
3. J. Gawlick and T. Hamacher, “Impact of coupling the electricity and hydrogen sector in a zero-emission European energy system in 2050,” *Energy Policy*, vol. 180, p. 113646, 2023.
4. Documentation of atlite: A Lightweight Python Package for Calculating Renewable Power Potentials and Time Series. atlite Developers . [Online]. Available: <https://atlite.readthedocs.io/en/latest/>
5. S. Akar, P. Beiter, W. Cole, D. Feldman, P. Kurup, E. Lantz, R. Margolis, D. Oladosu, T. Stehly, G. Rhodes, C. Turchi, and L. Vimmerstedt, “2020 Annual Technology Baseline (ATB) Cost and Performance Data for Electricity Generation Technologies,” National Renewable Energy Laboratory, Golden, CO, Tech. Rep., 2020, last updated: December 18, 2024. [Online]. Available: <https://doi.org/10.7799/1644189>
6. E. K. Gøtske, G. B. Andresen, F. Neumann, and M. Victoria, “Designing a sector-coupled European energy system robust to 60 years of historical weather data,” *Nature Communications*, vol. 15, no. 1, pp. 1–12, 2024.
7. F. Neumann, E. Zeyen, M. Victoria, and T. Brown, “The potential role of a hydrogen network in Europe,” *Joule*, vol. 7, no. 8, pp. 1793–1817, 2023.
8. D. Bogdanov, J. Farfan, K. Sadovskaia, A. Aghahosseini, M. Child, A. Gulagi, A. S. Oyewo, L. de Souza Noel Simas Barbosa, and C. Breyer, “Radical transformation pathway towards sustainable electricity via evolutionary steps,” *Nature communications*, vol. 10, no. 1, pp. 1–16, 2019.
9. C. Breyer, D. Bogdanov, M. Ram, S. Khalili, E. Vartiainen, D. Moser, E. Roman Medina, G. Masson, A. Aghahosseini, T. N. Mensah et al., “Reflecting the energy transition from a European perspective and in the global context—relevance of solar photovoltaics benchmarking two ambitious scenarios,” *Progress in Photovoltaics: Research and Applications*, vol. 31, no. 12, pp. 1369–1395, 2023.
10. C. Perpina Castillo, C. Hormigos Feliu, C. Dorati, G. Kakoulaki, L. Peeters, E. Quaranta, N. Taylor, A. Uihlein, D. Auteri, and L. Dijkstra, *Renewable Energy Production and Potential in EU Rural Areas*. Luxembourg: Publications Office of the European Union, 2024, no. JRC135612.
11. B. Willockx, C. Lavaert, and J. Cappelle, “Performance evaluation of vertical bifacial and single-axis tracked agrivoltaic systems on arable land,” *Renewable Energy*, vol. 217, p. 119181, 2023 .
12. “Renewable Power Generation Costs in 2023,” International Renewable Energy Agency, Abu Dhabi, Tech. Rep., 2024, accessed: 2024-12-25. [Online]. Available: <https://www.irena.org/Publications/2024/Sep/Renewable-Power-Generation-Costs-in-2023>

Response to Reviewer #2:

The work focuses on the highly relevant issue of energy security and the energy transition in Europe, presenting an innovative approach to addressing the problem. The analysis emphasizes the feasibility of accelerating renewable energy deployment rates based on recent trends, despite existing challenges. This study makes a valuable contribution to energy system modeling by integrating emerging solar configurations and incorporating self-sufficiency constraints. It provides practical insights into balancing regional equity, cost-efficiency, and energy security.

Compared to established literature, this study extends previous analyses by incorporating more granular modeling (e.g., inverter dimensioning, regional PV configurations) and employing a dynamic approach to transitioning towards self-sufficiency. The conclusions are well-supported by robust data and modeling. The integration of historical trends with scenario analyses provides a solid foundation for the claims.

Thank you very much for taking the time to review our paper. Your suggestions were a great help to us in revising and improving the paper. We detail below our responses along with the revisions that were carried out based on each comment.

Suggestions for improvement:

1. Detailed datasets: The datasets for specific assumptions, such as cost reductions, could be elaborated in greater detail to ensure clarity and reproducibility.

Response: We have added a table to the supplementary materials detailing cost assumptions for some of the main technologies. The complete dataset of cost assumptions for each investment year is also now available in Zenodo:

Supplementary Tables

The table below presents projected costs for 2030 for some selected technologies. The complete set of technology data assumptions for all investment periods is available in the Zenodo repository of this study. It corresponds to the PyPSA technology-data v0.8.1 (<https://github.com/PyPSA/technology-data/releases>). As noted in Supplementary Note 3, the solar PV costs in this study are lower due to the implementation of inverter dimensioning.

Supplementary Table 4: Summary of the main technology assumptions

technology	parameter	value	unit	currenc year
solar-utility	AC share of investment	21.6	%	2020
	FOM	2.5	%/year	2020
	investment	320.9	EUR/kW_e	2020
	lifetime	40	years	2020
solar-hsat	AC share of investment	18.4	%	2020
	FOM	2.3	%/year	2020
	investment	377.5	EUR/kW_e	2020
	lifetime	40	years	2020
solar-rooftop	FOM	1.4	%/year	2020
	discount rate	0.04	per unit	2015
	investment	668.0	EUR/kW_e	2020
	lifetime	40	years	2020
onwind	FOM	1.2	%/year	2015
	VOM	1.4	EUR/MWh	2015
	investment	1095.8	EUR/kW	2015
	lifetime	30	years	2015
⋮				
electricity distribution grid	FOM	2	%/year	2015
	investment	529.1	EUR/kW	2015
	lifetime	40	years	2015
	FOM	2	%/year	2015
	investment	148.1	EUR/kW	2015
	lifetime	40	years	2015

2. Citation format: Citations should not be grouped together; instead, they should be presented individually to provide more precise references.’

Response: Many thanks for pointing out this issue. We only maintain those references grouped together that support a statement indicating the common practice in literature, for example :

“Solar PV electricity is highlighted as the most cost-effective mitigation investment globally¹⁴, and its deployment increases when pursuing regional equity or reducing gas imports for Europe^{9,15,16,17,18}. ”

3. Equation S2: The cost parameter appears to be missing. Including its value is essential for a complete understanding of the equation.

Response: We have revised the equation and the description to clarify how the costs are calculated. Since the cost of components changes for each investment year, the values are included in Supplementary Table 1 instead of the equation itself:

Supplementary Note 3: Cost and land-use calculations for alternative solar configurations

1. Inverter dimensioning

In a PV installation, one part of the capital cost depends on the DC capacity (e.g. PV modules, land-use costs, soft costs), represented in Supplementary Eq. (2) as

DC components capital cost $\frac{DC}{AC}=1$, while the other part depends on the AC capacity of the plant (e.g. inverter, grid-connection), represented as

AC components capital cost $\frac{DC}{AC}=1$ $\left(\frac{\text{€}}{MW_{AC}}\right)$. Therefore, for a PV power plant with a DC/AC ratio of r :

$$\begin{aligned}
 & \text{PV power plant capital cost}_{\frac{DC}{AC}=r} \left(\frac{\text{€}}{MW_{AC}}\right) = \\
 & \text{DC components capital cost}_{\frac{DC}{AC}=1} \left(\frac{\text{€}}{MW_{DC}}\right) \cdot r \left(\frac{MW_{DC}}{MW_{AC}}\right) + \\
 & \text{AC components capital cost}_{\frac{DC}{AC}=1} \left(\frac{\text{€}}{MW_{AC}}\right)
 \end{aligned}$$

To see the impact of the new calculation method, which incorporates inverter sizing, consider the following example: In previous studies using PyPSA-Eur, the investment cost for utility-scale PV in 2030 was assumed to be 380.4 €/kW, based on DEA. However, this cost corresponds to a module with a DC/AC ratio of 1.25. This detail can be easily overlooked, as pairing a 1 kW solar panel with a 0.8 kW inverter results in less than a 3% loss in annual solar generation. By applying Equation (2), the investment cost for utility-scale solar PV is recalculated to 320.8€/kW_{AC} (251.3€ for the DC components + €69.5€ for the AC components), effectively reducing the investment cost by 15.8% relative to a PV system with the same DC and AC capacity.

Detailed assumptions for DC and AC components' capital cost for different investment years are shown in Supplementary Table 1. The final capital cost for fixed panels and horizontal single-axis tracking (HSAT) with different inverter ratios as calculated by Supplementary Eq. (2) is also shown in Supplementary Table 1.

4. Figures S12 to S21: The small chart sections in these figures are not clearly visible due to the color or scale. Adjusting the scale or improving the color contrast would enhance readability.

Response: Thank you for pointing out this deficiency. We have revised all the figures (S12-S21) to increase clarity. Hopefully, now readers will be able to discern different technologies without difficulty. Examples of revised figures are shown below:

5. Methodology rigor: The methodology is rigorous, utilizing the PyPSA-Eur model to analyze sector-coupled systems at high spatiotemporal resolution. The inclusion of innovative solar PV configurations is a particularly commendable enhancement.

Response: Thank you very much for your kind comment. We are very glad the methodology is satisfactory. We have already included the horizontal solar tracking in the default PyPSA-Eur model and are planning to add inverter dimensioning as well, so that other researchers can also conduct analyse with these configurations.

6. Empirical validation: Additional empirical validation, especially for novel configurations such as delta PV, would strengthen the study’s findings and provide further credibility to the conclusions.

These improvements would significantly enhance the clarity, precision, and impact of the paper.

Response: Overall, we have validated the energy generation profiles for various solar configurations (Fixed, HSAT, and Delta) produced using our renewable modeling package (atlite) against the PVGIS database [1]. PVGIS is a web-based tool developed by the European Commission that uses irradiance data from several satellite and reanalysis sources. It incorporates a rigorous methodology and is widely regarded as a reliable tool for estimating solar PV generation in Europe. The validation of east and west-facing panel generation is presented in Supplementary Figure 7 for two locations.:

Supplementary Fig. 7: Hourly capacity factor for a day (average throughout the year or years) for the Berlin and Paris nodes: (left column) as calculated by the atlite package using radiation data from satellite-aided SARA-2 dataset and temperature data from ERA5 reanalysis dataset for weather year 2013, compared with values for the same locations in (middle column) 2013 and (right column) 2005 to 2020 from the SARA-2 dataset as calculated by PVGIS web interface. All figures indicate a higher PV production in the morning than in the evening, but the bias from atlite is higher than the PVGIS data for 2013. The higher production in morning hours was also seen in the results of Szabo et al. when investigating vertical bifacial PV systems in Europe [2]. Note: the timestamp variation between ERA5 data and SARA-2 dataset has been accounted for in the current study by shifting the SARA-2 data by -30 minutes. For more details refer to PVGIS documentation on what the timestamps in each dataset represent.

To expand on this, we have added comparisons of the annual energy generation of the Delta configuration from our model with results from PVGIS for 10 different locations. Additionally, we have referenced a study that confirm PVGIS's accuracy in estimating total energy generation, showing its consistency with results from a large rooftop installation featuring the Delta configuration [3]. We have also cited a recent paper that uses PVGIS to model vertical bifacial PV systems (essentially a Delta configuration with a 90-degree tilt), further confirming the tool’s reliability for modeling PV generation [2].

Supplementary Fig. 8: Comparison between total annual generation of East-facing and West-facing PV panels with 10° tilt calculated by atlite and using data from PVGIS for 10 locations in Europe, averaged over 18 years from 2005 to 2023. Furthermore, results from Alkan et. al [3] confirm that generation from PVGIS matches well with experimental results from a 170 kW_p rooftop delta PV installation located in Istanbul, Turkey.

This is of course not an empirical validation in the traditional sense, as it does not involve experimental data. We attempted to find reliable experimental studies to validate our results, but the main issue is that each study has very specific parameters (i.e. shading conditions, inverter connection, etc.) that make the ‘actual production’ specific to that experiment. Nevertheless, we include a comparison of normalized generation profiles for Fixed and HSAT configurations against a recent experimental study [4]. The study was conducted in the US, so no comparison can be done for annual generation. However, the strong agreement between the observed generation profile for a single day and the average monthly profile from a node in Spain (which shares a similar latitude to the experimental site in Douglas County, CO) confirms that our model captures reality with good accuracy.

Figure 1- Comparison between normalized generation profile for left) fixed south-facing PV panel and right) PV with HSAT. The experimental data is taken from Micheli et al. [4], showing results for the same location (Douglas County, CO) and on the same day (May 4th, 2018). The modeled profiles produced by atlite are average daily generation for the Valencia node in Spain (same latitude as Douglas County), for June 2013.

References:

1. Photovoltaic Geographical Information System (PVGIS). [Online]. Available: https://joint-research-centre.ec.europa.eu/photovoltaic-geographical-information-system-pvgis/getting-started-pvgis/pvgis-user-manual_en
2. L. Szabo, M. Moner-Girona, A. Jager-Waldau, I. Kougias, A. Mezosi, F. Fahl, and S. Szabo, "Impacts of large-scale deployment of vertical bifacial photovoltaics on European electricity market dynamics," *Nature Communications*, vol. 15, no. 1, p. 6681, 2024.
3. S. Alkan and Y. Ates, "Pilot scheme conceptual analysis of rooftop east–west oriented solar energy system with optimizer," *Energies*, vol. 16, no. 5, p. 2396, 2023.
4. L. Micheli, M. Muller, M. Theristis, G. P. Smestad, F. Almonacid, and E. F. Fernandez, "Quantifying the impact of inverter clipping on photovoltaic performance and soiling losses," *Renewable Energy*, vol. 225, p. 120317, 2024.

Response to Reviewer #1:

Thank you for your thorough revisions and detailed elaborations. Your sensitivity analyses in the supplementary information have effectively addressed my concerns, demonstrating a high level of transparency and a robust methodological approach.

Regarding "Comment 3" from my initial review, and based on your responses where you state that these limits are rather arbitrary: I believe the removal of the evaluating word "only" in line 395 could have been considered. However, this is a minor point, as your assumptions are already well contextualized in line 144.

Overall, all comments have been addressed in an exemplary manner and relevant information has been added to the manuscript.

In summary, this study makes a valuable contribution to the energy system modeling research community by integrating emerging solar configurations and exploring important aspects of system analysis, such as self-sufficiency constraints in a carbon-neutral Europe.

I am pleased to recommend this study for publication. Congratulations!

Response: To ensure a proper wording is used, we have now removed the word 'only' from line 395, where the assumptions for calculating solar PV potential are outlined.

We are very happy that the changes and clarifications made during the revision are satisfactory. Thank you again for taking the time to review our paper and for the valuable feedback that has helped us improve the manuscript for readers.

Response to Reviewer #2:

Thank you for considering the revisions. After reviewing the changes made by the author, I find the paper suitable for publication.

Response: We are very happy that the changes and clarifications made during the revision are satisfactory. Thank you again for taking the time to review our paper and for the valuable feedback that has helped us improve the manuscript for readers.